# Metabolic constraints drive self-organization of specialized cell groups

**Sriram Varahan[1], Adhish Walvekar[1], Vaibhhav Sinha[2,3], Sandeep Krishna[2], Sunil Laxman[1]***

[1]InStem - Institute for Stem Cell Science and Regenerative Medicine, Bangalore, India; [2]Simons Centre for the Study of Living Machines, National Centre for Biological Sciences-Tata Institute of Fundamental Research, Bangalore, India; [3]Manipal Academy of Higher Education, Manipal, India

**Abstract** How phenotypically distinct states in isogenic cell populations appear and stably co-exist remains unresolved. We find that within a mature, clonal yeast colony developing in low glucose, cells arrange into metabolically disparate cell groups. Using this system, we model and experimentally identify metabolic constraints sufficient to drive such self-assembly. Beginning in a uniformly gluconeogenic state, cells exhibiting a contrary, high pentose phosphate pathway activity state, spontaneously appear and proliferate, in a spatially constrained manner. Gluconeogenic cells in the colony produce and provide a resource, which we identify as trehalose. Above threshold concentrations of external trehalose, cells switch to the new metabolic state and proliferate. A self-organized system establishes, where cells in this new state are sustained by trehalose consumption, which thereby restrains other cells in the trehalose producing, gluconeogenic state. Our work suggests simple physico-chemical principles that determine how isogenic cells spontaneously self-organize into structured assemblies in complimentary, specialized states.

***For correspondence:**
sunil@instem.res.in

**Competing interests:** The authors declare that no competing interests exist.

## Introduction

During the course of development, groups of isogenic cells often form spatially organized, interdependent communities. The emergence of such phenotypically heterogeneous, spatially constrained sub-populations of cells is considered a requisite first step towards multicellularity. Here, clonal cells proliferate and differentiate into phenotypically distinct cells that stably coexist, and organize spatially with distinct patterns and shapes (*Newman, 2016*; *Niklas, 2014*). Through such collective behavior, groups of cells can maintain orientation, stay together, and specialize in different tasks through the division of labor, while remaining organized with intricate spatial arrangements (*Ackermann, 2015*; *Newman, 2016*). In both eukaryotic and prokaryotic microbes, such organization into structured, isogenic but phenotypically heterogeneous communities, is widely prevalent, and also reversible (*Ackermann, 2015*). Such phenotypic heterogeneity within groups of clonal cells enables several microbes to persist in fluctuating environments, thereby providing an adaptive benefit for the cell community (*Wolf et al., 2005*; *Thattai and van Oudenaarden, 2004*).

A well studied example of spatially organized, phenotypically heterogeneous groups of cells comes from the *Dictyostelid* social amoeba, which upon starvation transition from individual protists to collective cellular aggregates that go on to form slime-molds, or fruiting bodies (*Bonner, 1949*; *Du et al., 2015*; *Kaiser, 1986*). Indeed, most microbes show some such complex, heterogeneous cell behavior, for example in the extensive spatial organization within clonal bacterial biofilms and swarms (*Kearns et al., 2004*; *Kolter, 2007*), or in the individuality exhibited in *Escherichia coli* populations (*Spudich and Koshland, 1976*). Despite its popular perception as a unicellular microbe, natural isolates of the budding yeast, *Saccharomyces cerevisiae*, also form phenotypically heterogeneous, multicellular communities (*Cáp et al., 2012*; *Koschwanez et al., 2011*; *Palková and*

**eLife digest** Under certain conditions, single-celled microbes such as yeast and bacteria form communities of many cells. In some cases, the cells in these communities specialize to perform specific roles. By specializing, these cells may help the whole community to survive in difficult environments. These co-dependent communities have some similarities to how cells specialize and work together in larger living things – like animals or plants – that in some cases can contain trillions of cells.

Research has already identified the genes involved in creating communities from a population of identical cells. It is less clear how cells within these communities become specialized to different roles. The budding yeast *Saccharomyces cerevisiae* can help to reveal how genetic and environmental factors contribute to cell communities.

By growing yeast in conditions with a low level of glucose, Varahan et al. were able to form cell communities. The communities contained some specialized cells with a high level of activity in a biochemical system called the pentose phosphate pathway (PPP). This is unusual in low-glucose conditions. Further examination showed that many cells in the community produce a sugar called trehalose and, in parts of the community where trehalose levels are high, cells switch to the high PPP state and gain energy from processing trehalose.

These findings suggest that the availability of a specific nutrient (in this case, trehalose), which can be made by the cells themselves, is a sufficient signal to trigger specialization of cells. This shows how simple biochemistry can drive specialization and organization of cells. Certain infections are caused by cell communities called biofilms. These findings could also contribute to new approaches to preventing biofilms. This knowledge could in turn reveal how complex multi-cellular organisms evolved, and it may also be relevant to studies looking into the development of cancer.

*Váchová, 2016*; *Ratcliff et al., 2012*; *Váchová and Palková, 2018*; *Veelders et al., 2010*; *Wloch-Salamon et al., 2017*). However, despite striking descriptions on the nature and development of phenotypically heterogeneous states within groups of cells, the rules governing the emergence and maintenance of new phenotypic states within isogenic cell populations remain unclear.

Current studies emphasize genetic and epigenetic changes that are required to maintain phenotypic heterogeneity within a cell population (*Ackermann, 2015*; *Sneppen et al., 2015*). In particular, many studies emphasize stochastic gene expression changes that can drive phenotypic heterogeneity (*Süel et al., 2007*; *Ackermann, 2015*; *Balázsi et al., 2011*; *Blake et al., 2003*). Further, groups of cells can produce adhesion molecules to bring themselves together (*Halfmann et al., 2012*; *Halme et al., 2004*; *Octavio et al., 2009*; *Váchová and Palková, 2018*), or support possible co-dependencies (such as commensal or mutual dependencies on shared resources) within the populations (*Ackermann, 2015*). Such studies now provide insight into why such heterogeneous cell groups might exist, and what the evolutionary benefits might be. However, an underlying biochemical logic to explain how distinct, specialized cell states can emerge and persist in the first place is largely absent. This is particularly so for isogenic (and therefore putatively identical) groups of cells in seemingly uniform environments. In essence, are there simple chemical or physical constraints, derived from existing biochemical rules and limitations, that explain the emergence and maintenance of heterogeneous phenotypic states of groups of clonal cells in space and over time?

Contrastingly, a common theme occurs in nearly all described examples of phenotypically heterogeneous, isogenic groups of cells. This is a requirement of some 'metabolic stress' or nutrient limitation that is necessary for the emergence of phenotypic heterogeneity and spatial organization, typically in the form of metabolically inter-dependent cells (*Ackermann, 2015*; *Campbell et al., 2016*; *Cáp et al., 2012*; *Johnson et al., 2012*; *Liu et al., 2015*). This idea has been explored experimentally, where approaches that systems-engineer metabolic dependencies between non-isogenic cells can result in interdependent populations that constitute mixed communities (*Campbell et al., 2016*; *Campbell et al., 2015*; *Embree et al., 2015*; *Wintermute and Silver, 2010*). These findings suggest that biochemical constraints derived from metabolism may determine the nature of phenotypic heterogeneity, and the spatial organization of cells in distinct states within the population. Therefore, if we can understand what these biochemical constraints are, and discern how metabolic

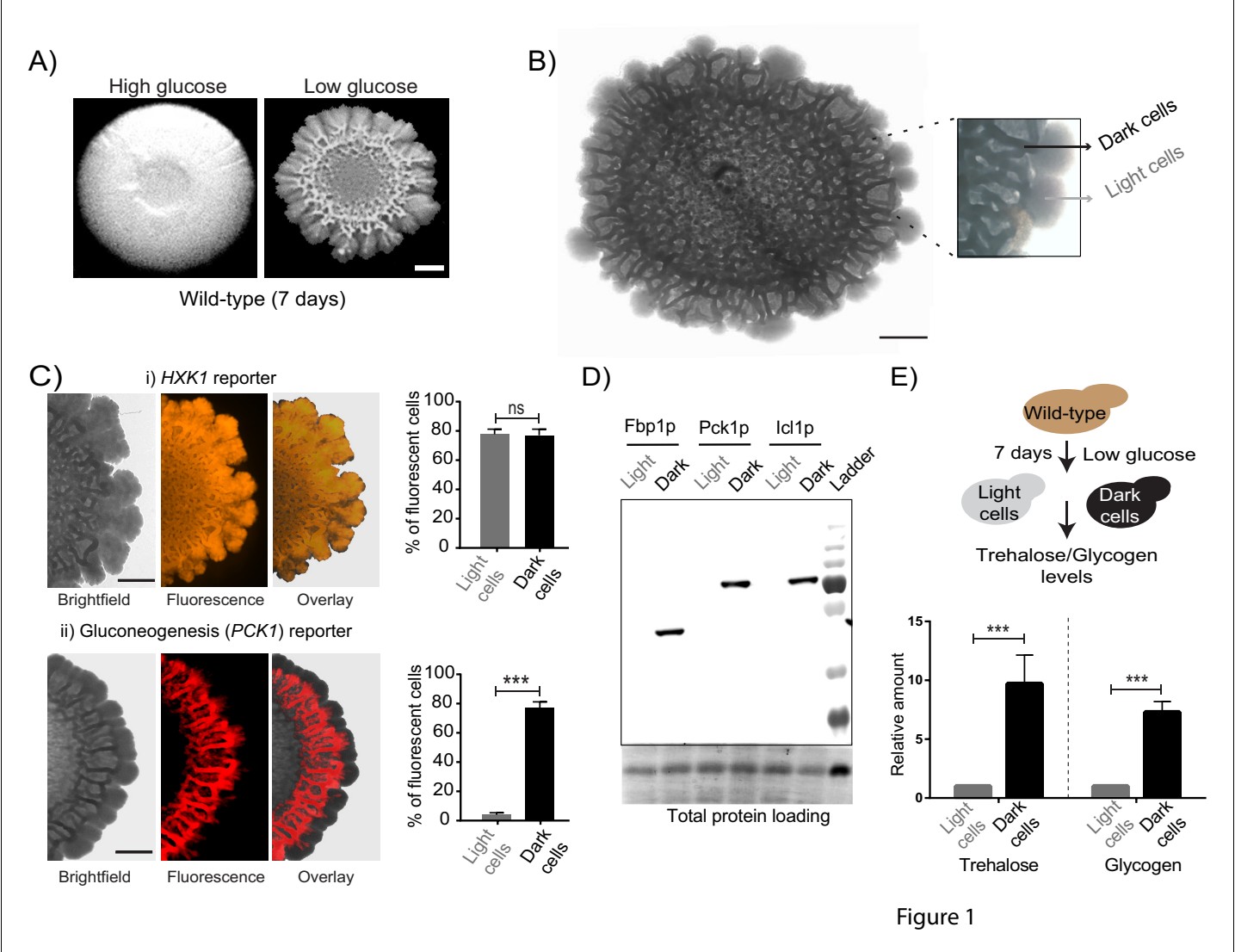

**Figure 1.** Cells within *S.cerevisiae* colonies exhibit ordered metabolic specialization. (**A**) Low glucose is required for rugose colonies to develop. The panel shows the morphology of mature yeast colonies in rich medium, with supplemented glucose as the sole variable. Scale bar: 2 mm. (**B**) Reconstructed bright-field images of a mature wild-type colony. Within the colony, a network of dark and bright regions is clearly visible, as classified based purely on optical density. We classify the cells in the dark region as dark cells, and in the peripheral light region as light cells. Scale bar: 2 mm. (**C**) Spatial distribution of mCherry fluorescence across a colony, indicating the activity of (i) a reporter for hexokinase (*HXK1*) activity, or (ii) a gluconeogenesis dependent reporter (*PCK1*), in two different colonies. The percentage of fluorescent cells (in isolated light and dark cells from the respective colonies) were also estimated by flow cytometry, and is shown as bar graphs. Scale bar: 2 mm. Also see *Figure 1—figure supplement 1A–B* and *Figure 1—figure supplement 2A* for more information. (**D**) Western blot based detection of proteins involved in gluconeogenesis (Fbp1p and Pck1p), or associated with increased gluconeogenic activity (Icl1p), in isolated dark or light cells. The blot is representative of three biological replicates (n = 3). Also see *Figure 1—figure supplement 2B* for more information. (**E**) Comparative steady-state amounts of trehalose and glycogen (as gluconeogenesis end point metabolites), in light and dark cells (n = 3). Statistical significance was calculated using unpaired t test (*** indicates p<0.001) and error bars represent standard deviation.

The online version of this article includes the following figure supplement(s) for figure 1:

**Figure supplement 1.** Gluconeogenesis activity is spatially restricted.
**Figure supplement 2.** Gluconeogenesis activity is spatially restricted.

states can be altered through these constraints, this may address how genetically identical cells can self-organize into distinct states.

In this study, using clonal yeast cells, we experimentally and theoretically show how metabolic constraints imposed on a population of isogenic cells can determine the production, accumulation,

and utilization of a specific, shared resource. The selective utilization of this resource enables the spontaneous emergence and persistence of cells exhibiting a counter-intuitive metabolic state, with spatial organization. These metabolic constraints create inherent threshold effects, enabling some cells to switch to new metabolic states, while restraining other cells to the original state which produces the resource. This thereby drives the overall self-organization of cells into specialized, spatially ordered communities. Finally, this group of spatially organized, metabolically distinct cells confer a collective growth advantage to the community of cells, rationalizing why such spatial self-organization of cells into distinct metabolic states benefits the cell community.

## Results

### Cells within *S. cerevisiae* colonies exhibit ordered metabolic specialization

Using a well-studied *S. cerevisiae* isolate as a model (*Reynolds and Fink, 2001*), we established a simple system to study the formation of a clonal colony with irregular morphology. On 2% agar plates containing a complex rich medium with low glucose concentrations, *S. cerevisiae* forms rugose colonies with distinct architecture, after ~5–6 days (*Figure 1A*). Such colonies do not form in the typical, high (1–2%) glucose medium used for yeast growth (*Figure 1A*). Thus, as previously well established (*Granek and Magwene, 2010*; *Reynolds and Fink, 2001*), glucose limitation (with other nutrients such as amino acids being non-limiting) drives this complex colony architecture formation. Currently, the description of such colonies is limited to this external rugose morphology, and does not describe the phenotypic states of cells and/or any spatial organization in the colony. With only such a description, as observed in *Figure 1A*, the mature colony surface has an internal circle and some radial streaks near the periphery. We carried out a more detailed observation of entire colonies under a microscopic bright-field (using a 4x lens). Here, we unexpectedly noticed what appeared to be distinct internal patterning, and apparent spatial organization of cells within the colony (*Figure 1B*). As categorized purely based on these observed differences in visual optical density ('dark' or 'light'), regions between the colony center and periphery had optically dense (dark) networks spanning the circumference of the colony, interspersed with optically rare regions. In contrast, the periphery of the mature colony appeared entirely light (*Figure 1B*). Based simply on these optical traits alone, we categorized cells present in these regions of the colony as dark cells and light cells (*Figure 1B*). At this point, our description is visual and qualitative, and does not imply any other difference in the cells in either region. However, this visual description is both robust and simple, and hence we use this nomenclature for the remainder of this manuscript.

Since these structured colonies form only in glucose-limited conditions, we hypothesized that dissecting the expected metabolic requirements during glucose limitation might reveal drivers of this internal organization within the colony. The expected metabolic requirements of cells growing in glucose limited conditions are as follows: first, all cells would be expected to have constitutively high expression of the high-affinity hexokinase (Hxk1p) (*Lobo and Maitra, 1977*; *Rodríguez et al., 2001*). Further, during glucose-limited growth, all cells are expected to carry out extensive gluconeogenesis, as the default metabolic state (*Broach, 2012*; *Haarasilta and Oura, 1975*; *Yin et al., 2000*). Indeed, we confirmed this second expectation by measuring the amounts of the gluconeogenic enzymes Pck1 (phosphoenolpyruvate carboxykinase) and Fbp1 (fructose-1,6-bisphosphatase), in short-term (4–5 hr) liquid cultures of log-phase cells growing in either high glucose medium, or in the same glucose-limited medium we used for colony growth. Expectedly, we observed very high amounts of these gluconeogenic enzymes in cells growing in glucose-limited medium (*Figure 1—figure supplement 1A*), reiterating that even in well-mixed glucose-limited, cells are in a strongly gluconeogenic state. In order to now examine the mature colony and dissecting expected metabolic requirements in these conditions, we first designed visual indicators for these metabolic hallmarks of yeast cell growth in low glucose. We engineered two different fluorescent reporters, one dependent on *HXK1* expression (mCherry under the *HXK1* gene promoter), and the second on *PCK1* expression as an indicator of gluconeogenic activity (mCherry under the *PCK1* gene promoter) (*Figure 1—figure supplement 1B*). Cells carrying these reporters were seeded to develop into colonies, and the expression levels of these reporters were monitored in the mature, rugose colony (5–6 days). Expectedly, the *HXK1*-promoter dependent reporter showed constitutive, high expression in all cells across

**Table 1.** Mass transitions used for LC-MS/MS experiments.

| Nucleotides | Formula | Parent/Product (positive polarity) | Comment (for 15N experiment) |
|---|---|---|---|
| AMP | $C_{10}H_{14}N_5O_7P$ | 348/136 | Product has all N |
| 15N_AMP_1 | | 349/137 | |
| 15N_AMP_2 | | 350/138 | |
| 15N_AMP_3 | | 351/139 | |
| 15N_AMP_4 | | 352/140 | |
| 15N_AMP_5 | | 353/141 | |
| GMP | $C_{10}H_{14}N_5O_8P$ | 364/152 | Product has all N |
| 15N_GMP_1 | | 365/153 | |
| 15N_GMP_2 | | 366/154 | |
| 15N_GMP_3 | | 367/155 | |
| 15N_GMP_4 | | 368/156 | |
| 15N_GMP_5 | | 369/157 | |
| CMP | $C_9H_{14}N_3O_8P$ | 324/112 | Product has all N |
| 15N_CMP_1 | | 325/113 | |
| 15N_CMP_2 | | 326/114 | |
| 15N_CMP_3 | | 327/115 | |
| UMP | $C_9H_{13}N_2O_9P$ | 325/113 | Product has all N |
| 15N_UMP_1 | | 326/114 | |
| 15N_UMP_2 | | 327/115 | |
| **Trehalose and sugar phosphates** | **Formula** | **Parent/Product (negative polarity)** | **Comment (for 13C experiment)** |
| Trehalose | $C_{12}H_{22}O_{11}$ | 341.3/179.3 | |
| 13C_Trehalose_12 | | 353.3/185.3 | Product has 6 C all of which are labeled |
| G3P | $C_3H_7O_6P$ | 169/97 | Monitoring the phosphate release |
| 13C_G3P_3 | | 172/97 | |
| 3 PG | $C_3H_7O_7P$ | 185/97 | Monitoring the phosphate release |
| 13C_3 PG_3 | | 188/97 | |
| G6P | $C_6H_{13}O_9P$ | 259/97 | Monitoring the phosphate release |
| 13C_G6P_6 | | 265/97 | |
| 6 PG | $C_6H_{13}O_{10}P$ | 275/97 | Monitoring the phosphate release |
| 13C_6 PG_6 | | 281/97 | |
| R5P | $C_5H_{11}O_8P$ | 229/97 | Monitoring the phosphate release |
| 13C_R5P_5 | | 234/97 | |
| S7P | $C_7H_{15}O_{10}P$ | 289/97 | Monitoring the phosphate release |
| 13C_S7P_5 | | 294/97 | |

the entire colony (*Figure 1C*). Contrastingly, only the dark cells exhibited high gluconeogenesis reporter activity (*Figure 1C*). Notably, the light cells entirely lacked detectable gluconeogenic reporter activity (*Figure 1C*). To better quantify this phenomenon, cells were dissected out from dark or light regions respectively (under the light microscope, using a fine needle), and the percentage of fluorescent cells in each region was measured using flow cytometry. Based on flow cytometric readouts,~80% of the isolated dark cells showed strong fluorescence for the gluconeogenic reporter, while ~97% of the light cells were non-fluorescent for gluconeogenic activity (*Figure 1C*, *Figure 1— figure supplement 1C*). This spatial distribution of gluconeogenic activity is shown as a quantitative heat-map histogram overlaid on the entire colony, in *Figure 1—figure supplement 2A*. Since this

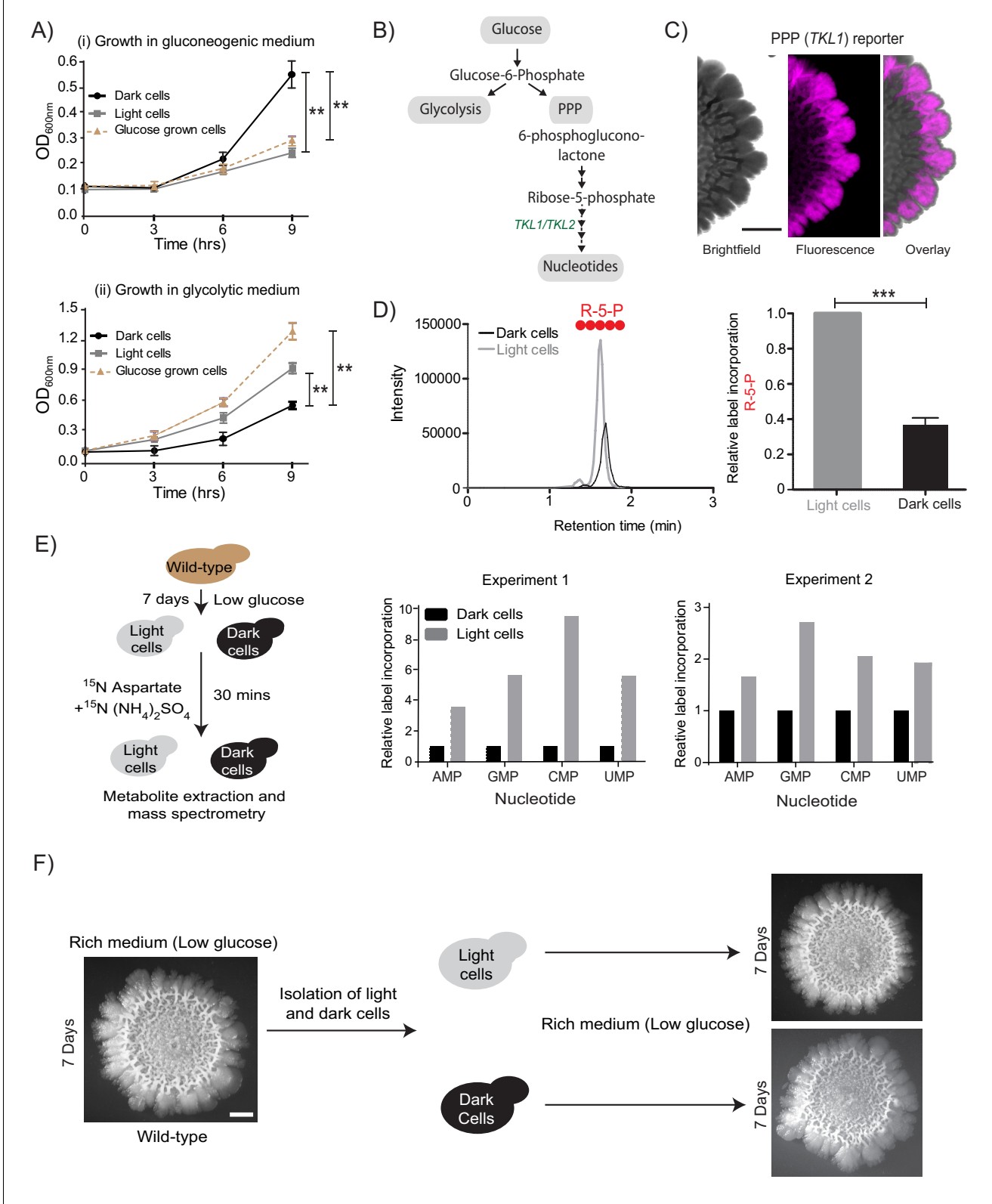

**Figure 2.** Cells organize into spatially restricted, contrary metabolic states within the colony. (**A**) Comparative immediate growth of isolated light cells and dark cells, transferred to a 'gluconeogenic medium' (2% ethanol as carbon source), or a 'glycolytic medium' (2% glucose as carbon source), based on increased absorbance ($OD_{600}$) in culture. Wild-type cells growing in liquid medium (2% glucose) in log phase (i.e. in a glycolytic state) were used as controls for growth comparison (n = 3). (**B**) A schematic showing metabolic flow in glycolysis and the pentose phosphate pathway (PPP), and also

*Figure 2 continued on next page*

*Figure 2 continued*

illustrating the synthesis of nucleotides (dependent upon pentose phosphate pathway). *TKL1* controls an important step in the PPP, and is strongly induced during high PPP flux. (C) Spatial distribution of mCherry fluorescence across a colony, based on the activity of a PPP- dependent reporter. Scale bar: 2 mm. Also see *Figure 1—figure supplement 1A* and *Figure 2—figure supplement 1A*. (D) LC-MS/MS based metabolite analysis, using exogenously added $^{13}$C Glucose, to compare flux of $^{13}$C Glucose into the PPP metabolite ribulose-5-phosphate (R-5-P), in light and dark cells. The red circles represent $^{13}$C labeled carbon atoms (n = 3). Also see *Figure 2—figure supplement 1B*. (E) Comparative metabolic-flux based analysis comparing $^{15}$N incorporation into newly synthesized nucleotides, in dark and light cells. Also see S2, and Materials and methods. (F) Light cells and dark cells isolated from a 7 day old wild-type complex colony re-form indistinguishable mature colonies when re-seeded onto fresh agar plates, and allowed to develop for 7 days. Scale bar = 2 mm. Statistical significance was calculated using unpaired t test (*** indicates p<0.001, ** indicates p<0.01) and error bars represent standard deviation.

The online version of this article includes the following figure supplement(s) for figure 2:

**Figure supplement 1.** Light cells exhibit high PPP activity.

observation was based solely on reporter activity, in order to more directly examine this observation, we estimated native protein amounts of enzymes associated with gluconeogenesis (Pck1, Fbp1, and Icl1- Isocitrate lyase from the glyoxylate shunt) in isolated light cells and dark cells. Only the dark cells showed expression of the gluconeogenic enzymes (*Figure 1D*, *Figure 1—figure supplement 2B*). Finally, we measured steady-state amounts of trehalose and glycogen within dark and light cells, using these metabolites as unambiguous biochemical readouts of the end-point biochemical outputs of gluconeogenesis (*François et al., 1991*). We observed that the dark cells had substantially higher amounts of both trehalose and glycogen (*Figure 1E*), indicating greater gluconeogenic activity in these cells. Collectively, these results strikingly reveal that intracellular gluconeogenic activity is spatially restricted to specific regions, resulting in a distinct pattern of metabolically specialized zones within the colony.

## Cells organize into spatially restricted, contrary metabolic states within the colony

In the given nutrient conditions of low glucose, gluconeogenesis is an expected, constitutive metabolic process, essential for cells. This can therefore be considered as a necessary, permitted metabolic state in this condition. Paradoxically, in these mature colonies, gluconeogenic activity was spatially restricted to only within the dark cell region, with no discernible gluconeogenic activity in the cells located in the light region. This absence of gluconeogenic activity in these light cells, while concomitant with a constitutively high level of hexokinase activity, therefore poses a biochemical paradox. What might the metabolic state of these light cells be? To quickly address this using a crude but useful readout, we compared the ability of freshly isolated light and dark cells to proliferate in both gluconeogenic (low glucose), and non-gluconeogenic (high glucose) growth conditions. For simplicity, isolated light cells and dark cells were inoculated either into a medium where gluconeogenesis is essential (2% ethanol +glycerol as a sole carbon source), or in high (2%) glucose medium where cells rely on high glycolytic and pentose phosphate pathway (PPP) activity, and initial cell proliferation was monitored. Here, cells that had been growing in high glucose were used as a control. Expectedly, the dark cells grew robustly and reached significantly higher cell numbers ($OD_{600}$) compared to the light cells in the gluconeogenic condition (*Figure 2A*). Conversely, light cells grew robustly when transferred to the high glucose medium, as compared to the dark cells (*Figure 2A*). While this was an overly simple, and not definitive experiment, counter-intuitively, this result suggested that despite being in a low-glucose environment, the light cells were well suited for growth in high glucose, and therefore might be in a metabolic state suited for growth in glucose. We therefore decided to more systematically investigate this phenomenon.

In the presence of glucose, yeast cells typically show high glycolytic and PPP activities, as part of the Crabtree (analogous to the Warburg) effect (*Crabtree, 1929*; *De Deken, 1966*; *Figure 2B*). Therefore, if the light cells in the colony were indeed behaving as though present in more glucose-replete conditions, they should exhibit high PPP activity. To test this, we first designed a fluorescent PPP-activity reporter (mCherry under the control of the transketolase 1 (*TKL1*) (*Walfridsson et al., 1995*) gene promoter, *Figure 1—figure supplement 1B*), and monitored reporter activity across the mature colony. Indeed, only the light cells exhibited high PPP-reporter activity (*Figure 2C*, *Figure 2—figure supplement 1A*). This spatial restriction of high PPP activity across the colony is also

**Table 2.** Strains and plasmids used in this study.

| Strain/genotype | Information | Source/reference |
|---|---|---|
| Wild-type (WT) | YBC16G1, prototrophic sigma1278b, *MAT a* | Isolate *via* Fink Lab |
| WT (*pPCK1-mCherry*) | *Wild-type strain with gluconeogenesis reporter plasmid (mCherry with PCK1 promoter)* | this study |
| WT (*pHXK1-mCherry*) | *Wild-type strain with constitutive reporter plasmid (mCherry with HXK1 promoter)* | this study |
| WT (*pTKL1-mCherry*) | *Wild-type strain with pentose phosphate pathway reporter plasmid (mCherry with TKL1 promoter)* | this study |
| *PCK1-flag* | *MAT a PCK1-3xFLAG::natNT2* | this study |
| *FBP1-flag* | *MAT a FBP1-3xFLAG::natNT2* | this study |
| *ICL1-flag* | *MAT a ICL1-3xFLAG::natNT2* | this study |
| *MAL11-flag* | *MAT a MAL11-3xFLAG::natNT2* | this study |
| *NTH1-flag* | *MAT a NTH1-3xFLAG::natNT2* | this study |
| Δnth1 | *MAT a nth1::kanMX6* | this study |
| Δmal11 | *MAT a mal11::kanMX6* | this study |
| Δnth1 (*pPCK1-mCherry*) | Δnth1 strain with gluconeogenesis reporter plasmid (mCherry with PCK1 promoter) | this study |
| Δmal11 (*pPCK1-mCherry*) | Δmal11 strain with gluconeogenesis reporter plasmid (mCherry with PCK1 promoter) | this study |
| Δnth1 (*pTKL1-mCherry*) | Δnth1 strain with pentose phosphate pathway reporter plasmid (mCherry with TKL1 promoter) | this study |
| Δmal11 (*pTKL1-mCherry*) | Δmal11 strain with pentose phosphate pathway reporter plasmid (mCherry with TKL1 promoter) | this study |
| Plasmid | Information | Source/reference |
| pPCK1-mCherry | *mCherry under the PCK1 promoter and CYC1 termin- ator. p417 centromeric plasmid backbone, $G418^R$.* | this study |
| pHXK1-mCherry | *mCherry under the HXK1 promoter and CYC1 termin- ator. p417 centromeric plasmid backbone, $G418^R$.* | this study |
| pTKL1-mCherry | *mCherry under the TKL1 promoter and CYC1 termin- ator. p417 centromeric plasmid backbone, $G418^R$.* | this study |

shown as an overlaid quantitative heat-map histogram in *Figure 2—figure supplement 1A*. Next, we directly addressed the possibility of these light cells exhibiting relatively high PPP activity. For unambiguously testing this, we utilized a stable-isotope based metabolic flux experiment to assess the flux towards PPP in light and dark cells. Light and dark cells isolated from colonies were pulsed with [13]C-labeled glucose (for ~5 min), metabolites extracted, and the incorporation of this carbon label into the late PPP intermediates ribulose-5-phosphate (R-5-P) and sedoheptulose-7-phosphate (S-7-P) was measured by liquid chromatography/mass spectrometry (LC/MS/MS). The relative amounts of these all-carbon labeled PPP intermediates were compared between the two cell types (light or dark). Notably, light cells incorporated significantly higher levels of [13]C labeled glucose into PPP metabolites compared to the dark cells (*Figure 2D*, and *Figure 2—figure supplement 1B*, and see *Table 1* for MS parameters), showing that the light cells are in a high PPP activity state. Finally, we assessed if other biochemical end-point outputs requiring high PPP activity/flux were also high in the light cells. High nucleotide synthesis is a canonical consequence of enhanced PPP activity

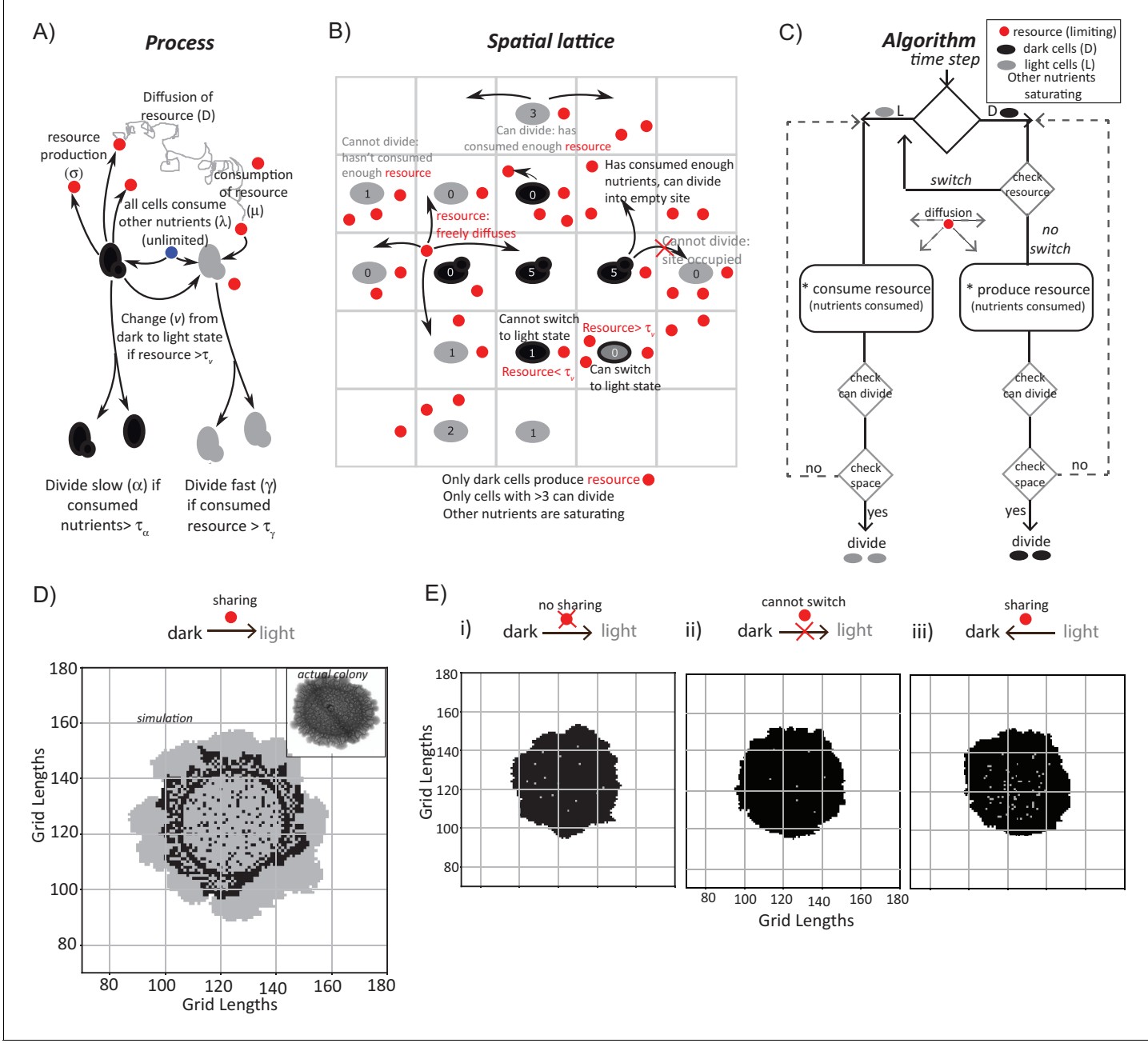

**Figure 3.** A mathematical model suggests constraints for the emergence and organization of cells in complimentary metabolic states. (**A**) Processes, based on experimental data, incorporated into developing a simple mathematical model to simulate colony development. The dark and light cells are appropriately colored, and the parameters incorporated are resource production (σ), diffusion parameters for the resource (D), consumption of the resource (μ), and fast or slow rates of division (α or γ), based on resource or amino acid consumption. (**B**) The spatial distribution of cells is reduced to a grid like lattice within the model, to allow coarse graining of the location of cells across a colony. The rules for cell division and expansion incorporate the ability to consume existing nutrients in the medium, produce a resource and/or consume a produced resource, and a threshold amount of resource build up before utilization. (**C**) A flow-chart of the algorithm used in the mathematical model. The decision making process in the algorithm, incorporating all the elements described in panels (**A**) and (**B**) is illustrated. Also see *Figure 3—figure supplement 1A–C* and Materials and methods. (**D**) A simulation of the development of a wild-type colony, based on the default model developed. The inset shows an image of a real wild-type colony (same brightfield image used in *Figure 1B*), which has developed for an equivalent time (~6 days). Also see *Figure 3—figure supplement 1A–C* and *Video 1*. (**E**) A simulation of colony development using the model, where key parameters have been altered. (**i**) The sharing of a produced resource is restricted. (**ii**) The ability to switch from a dark to a light state is restricted. (**iii**) Light cells produce a resource taken up by dark cells is included. Note that in all three scenarios the colony size remains small, and fairly static. Also see *Figure 3—figure supplement 2A–D* and *Videos 2*, *3* and *4*.

The online version of this article includes the following figure supplement(s) for figure 3:

*Figure 3 continued on next page*

*Figure 3 continued*

**Figure supplement 1.** Effects on the colony as we change individual parameters used in the model.
**Figure supplement 2.** Reproducibility of the model, under different scenarios.

(*Nelson and Cox, 2013*). The carbon backbone (ribose-5-phosphate) of newly synthesized nucleotides is derived from the PPP, while the nitrogen backbone comes from amino acids (*Nelson and Cox, 2013*; *Figure 2B*, and see *Table 1* for MS parameters). We devised another metabolic flux-based experiment to assess de novo nucleotide biosynthesis in light and dark cells, as an end-point collective readout of high PPP activity coupled with amino acid utilization. Light and dark cells, isolated from colonies were pulsed with a $^{15}$N-label (ammonium sulfate +aspartate), and incorporation of this label into nucleotides was measured by liquid chromatography/mass spectrometry (LC/MS/MS). Light cells had higher flux into nucleotide biosynthesis, compared to the dark cells (*Figure 2E*, and see *Table 1* for MS parameters). Taken together, we surprisingly find that light cells exhibit multiple metabolic hallmarks of cells growing in glucose-replete conditions, including increased PPP activity, and increased nucleotide biosynthesis. Thus, in the spatially organized colony, the light cells and dark cells have contrary metabolic states. This is despite the expectation that the gluconeogenic state, exhibited by the dark cells, is the plausible metabolic state in the given growth conditions.

Notably, the light cells or dark cells, when isolated and reseeded as a new colony, both develop into indistinguishable, complex colonies (*Figure 2F*). This reiterates that these phenotypic differences between the light and dark cells are fully reversible, and do not require genetic changes. Collectively, these data reveal that cells within the colony organize into spatially separated, metabolically specialized regions. Within these regions, cells exhibit complimentary metabolic states. One of these states, where cells have high PPP activity, is counter-intuitive and cannot obviously be sustained given the external nutrient environment.

## A mathematical model suggests constraints for the emergence and organization of cells in complimentary metabolic states

What determines the emergence and spatial organization of a group of cells, in these contrary metabolic states? Particularly, what can explain the emergence and proliferation of the light cells, which exhibit this counter-intuitive metabolic state, while the colony maintains a large subset of cells in the dark state? To address this, we built a coarse-grained mathematical model. This model incorporates

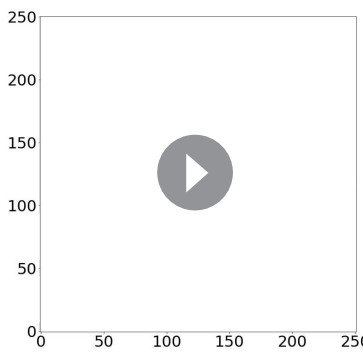

**Video 1.** Development of the colony. Simulation video showing the changes in a wild-type model colony. After a small lag, dark cells at the edge start dividing into empty space and due to the threshold switching effect, the center of the colony now has light cells. Cells at the periphery also switch to light cells which divide faster by utilizing resource shared by the dark cells.
https://elifesciences.org/articles/46735#video1

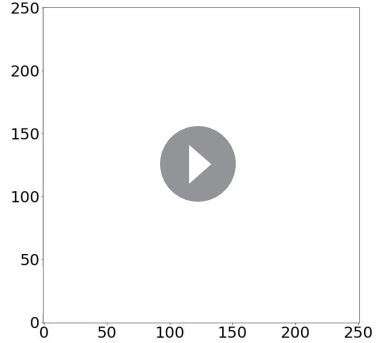

**Video 2.** Dark cells do not share the metabolic resource with light cells. Simulation video showing changes in a colony where the dark cells don't share any resource for the light cells to consume. The number of light cells doesn't increase and the final colonies predominantly comprise of dark cells.
https://elifesciences.org/articles/46735#video2

**Table 3.** Parameters of the model for the wild-type case.

| Parameter | Notation | Value |
|---|---|---|
| Growth rate (dark cell block) | $g_d$ | 0.01/T |
| Growth rate (light cell block) | $g_l$ | 0.04/T |
| Switching threshold | S | 3.0 units |
| Resource produced by each dark cell block | R | 0.07 units/T |
| Resource or amino acids consumed per cell block (light or dark) | C | 0.05 units/T |
| Minimum resource or amino acid reserve needed for division (light or dark) | – | 1.0 unit |
| Chance to switch to light cell if threshold reached | P | 0.5/T |
| Diffusion constant of the resource | D | 0.24 $L^2$/T |

simple processes derived from our current experimental data, to simulate the formation of a colony of 'light' and 'dark' cells. The model was intentionally coarse-grained, since its purpose was only to find a minimal, biologically consistent combination of processes that is sufficient to produce the overall spatial structure and composition of cell states observed in the colonies. The intention behind the model was not to decipher all possible molecular details that explain this phenomenon. The model should only sufficiently account for both the emergence of light cells, as well as their spatial organization with dark cells. Such a model could therefore suggest constraints that determine the emergence of light cells, and the organization of the colony with the observed organization, which can then be experimentally tested.

While building this model, we included a range of processes that must be considered, based on our experimental data thus far (*Figure 3A*). This includes (i) the dark cells switching to a light state, (ii) the production of some resource by dark cells, which may be shared/utilized by the cells, (iii) diffusion parameters for this resource, (iv) consumption of this resource, and (v) rates of cell division are included (*Figure 3A*). Next, we constructed a two-dimensional square grid of 'locations' for groups of cells within the colony (*Figure 3B*). Here, each location is either empty or occupied by a group of ~100 cells (also see Materials and methods for full details). Note: we intentionally coarse-grain the grid (for computational simplicity, in order to simulate colony sizes comparable to real colonies) by approximating that the locations either consist of all light or all dark cells. This is a major simplification that was necessary. At each time step (12 min of real time), all the processes shown in *Figure 3A* are executed across the spatial grid using the outlined algorithm (*Figure 3C*). In such an implemented algorithm, (i) all cells consume all available nutrients (present in saturating amounts), while free glucose concentrations are negligible, (ii) dark cells grow and divide in the given conditions, (iii) dark cells produce a resource/resources as a consequence of their existing metabolic (gluconeogenic) state, (iv) this resource diffuses around the grid and is freely available, (v) dark cells switch to the light state if sufficient resource is present at their location, and lastly, (vi) the resource when consumed can sustain the light state cells, which can expand if there is an empty location in the neighborhood. If the resource is not present in that location the light cells will switch back to dark cells. All processes occur at specified rates, allowing for stochasticity. Finally, this existence of a shared resource is surmised because, logically the emergence of light cells from dark can happen only if the local nutrient environment enables a switch to the new metabolic state.

In each simulation, empty grids are seeded with 1257 occupied locations, with 95–99% of the cells in the dark state. After ~750 time steps (corresponding to ~6 days) a simulated wild-type colony looks typically as shown in *Figure 3D* (also see *Video 1*). A range of resource amounts, growth rates

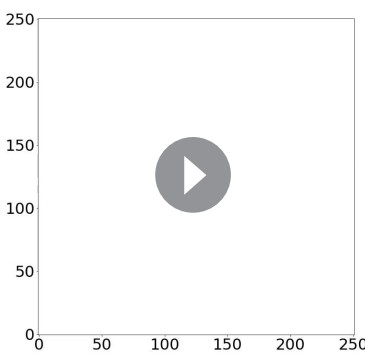

**Video 3.** Dark cells do not switch to light cells. Simulation video showing changes in a colony where the dark cells don't switch to light cells but continue to produce resource on to the resource grid. In certain cases, there can be light cells at the very edge of the starting colony. This is because the composition of the colonies might be the same between simulations, but individual cell block locations are done at random. Such cells at the edge can utilize the shared resource and divide into empty space on the grid.
https://elifesciences.org/articles/46735#video3

and diffusion of the resource were included in control simulations (see *Figure 3—figure supplement 1A–C*). Strikingly, the simulated spatial organization (*Figure 3D*, and *Video 1*) recapitulates most obvious features of a real colony (*Figure 3D*). These are: at the edge of the initial circular inoculation of the colony is a ring of dark cells, the outermost part of the colony is made up of outcrops of light cells, and from this ring of dark cells emanate clusters of dark cells penetrating into the outcrops of light cells. This is despite the simplicity of the rules in the model, including its flattening into 2D. In the simulation, for the first 40–45 time steps, the colony remains small and predominantly dark, while the resource builds up. Then, dark cells start to switch to light. When this happens within the bulk of the colony, these light cells have restricted division due to spatial constraints. Around 100 to 150 time steps later, light cells emerge at the perimeter of the colony, and then rapidly divide and expand (*Figure 3D*, *Video 1*). In order to test if the processes of *Figure 3A* are all required for this behavior, we examined three comprehensive control scenarios: (i) dark cells do not produce a resource (and therefore in this case for growth light cells depend only on amino acids or other pre-supplied resources in the medium), (ii) dark cells cannot switch to the light state, or (iii) light cells produce a resource that is needed by dark cells to grow (a straw-man scenario, since initially in the actual colony all cells were in a dark state, as shown earlier). None of these cases produces the wild-type spatial organization, over a wide range of parameter values (*Figure 3E*, as well as a range of parameters explored in *Figure 3—figure supplement 2A–D*, and simulations in *Videos 2–4*).

Summarizing, this simple model successfully recaptures the general features of the spatial patterning and organization of real colonies. This includes the overall general architecture, and spatial organization of light and dark cells. Two simple take-home points emerge from this model, for such spatial distribution of cells in these two metabolic states, across the developing colony. First, the model requires that dark (gluconeogenic) cells will produce a resource that is needed by dark cells to switch to the light state. Second, a resource produced by the dark cells is required to sustain the light state. Collectively, in our model, these metabolic constraints are sufficient to determine the overall spatial organization of metabolically distinct, specialized cells.

## Trehalose satisfies criteria to be the metabolic resource determining the emergence of light cells

Does any gluconeogenic metabolite(s) determine the organization of these cells, consistent with

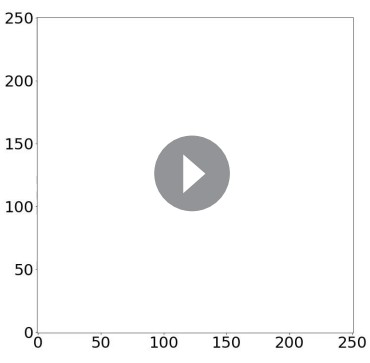

**Video 4.** Light cells share metabolic resource with dark cells (Wrong sharing). Simulation video showing changes in a colony where the dark cells don't share any resource for the light cells but the light cells provide amino acids for the dark cells to consume (wrong sharing). The dark cells have an abundance of amino acids to grow and divide. The final colonies predominantly comprise of dark cells.
https://elifesciences.org/articles/46735#video4

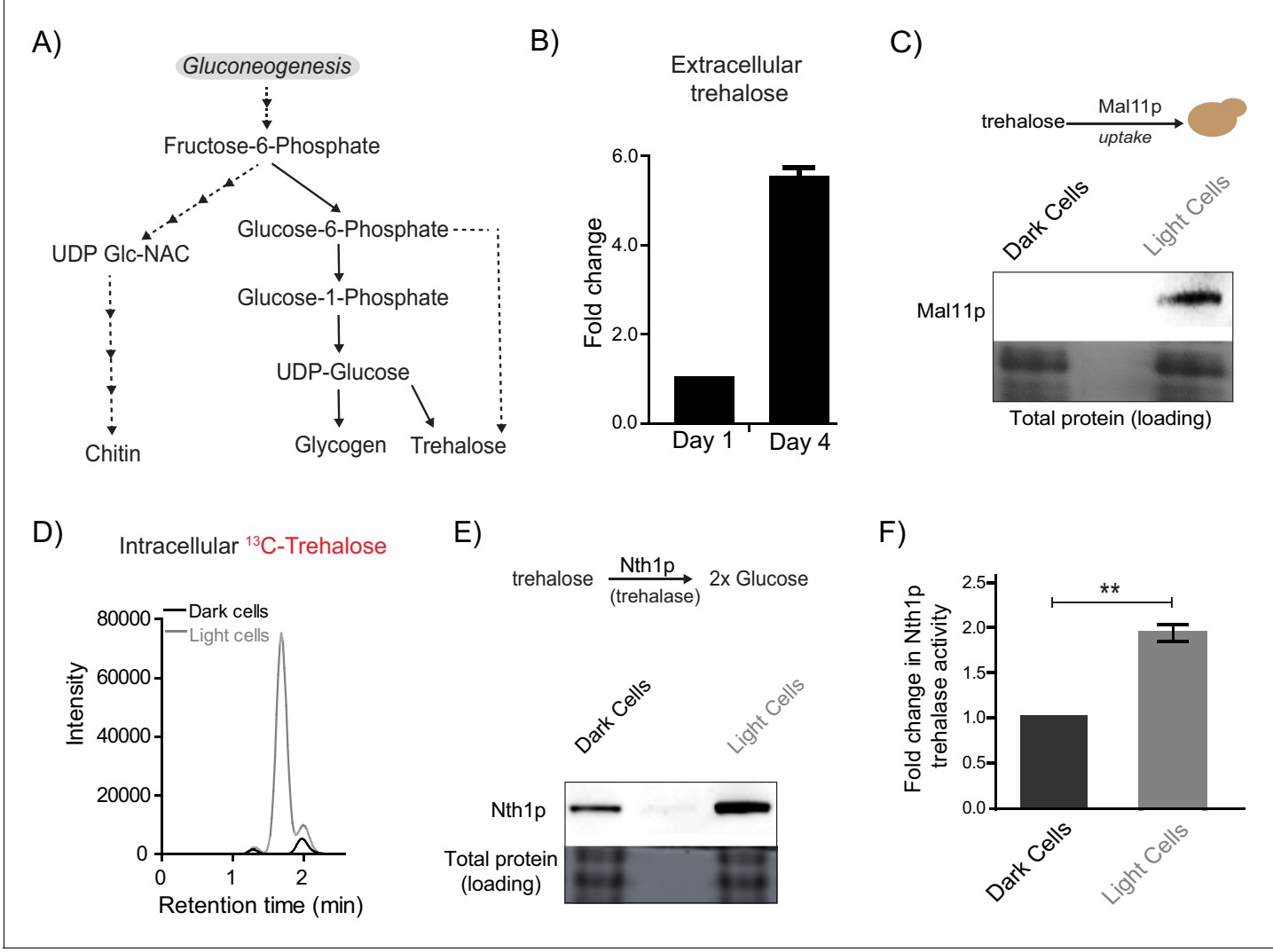

**Figure 4.** Trehalose satisfies criteria to be the metabolic resource determining the emergence of light cells. (**A**) A schematic illustrating the metabolic intermediates and different end-point metabolites of gluconeogenesis. (**B**) Extracellular amounts of trehalose measured from developing wild-type colonies. Entire colonies were isolated, and only exogenous trehalose estimated, at the respective days. Fold change in the amount of extracellular trehalose produced by a 4 day old colony with respect to a 1 day old colony was calculated (n = 3). (**C**) Comparative protein amounts of Mal11, a major transporter of trehalose in *S. cerevisiae*, in light and dark cells, as measured using a Western blot is shown. The blot is representative of 3 independent experiments (n = 3). (**D**) Estimates of the relative ability of light and dark cells to uptake trehalose is shown. $^{13}$C Trehalose was exogenously added to light and dark cells, and intracellular amounts of the same are shown (as intensity of the MS/MS peak corresponding to $^{13}$C-trehalose) (n = 3). (**E**) Comparative amounts of Nth1, the major intracellular trehalase enzyme in *S. cerevisiae*, in light and dark cells, as measured using a Western blot is shown. The blot is representative of 3 independent experiments (n = 3). (**F**) in vitro neutral trehalase activity present in lysed light or dark cells is shown (n = 3). Statistical significance was calculated using unpaired t test (** indicates p<0.01) and error bars represent standard deviation.

The online version of this article includes the following figure supplement(s) for figure 4:

**Figure supplement 1.** Quantitation of relative Mal11 and Nth1 protein levels in light and dark cells.

these requirements suggested by experimental and modeled data? Such a metabolite must logically satisfy the following three criteria. First, this resource should be available in the extracellular environment (i.e. released by cells), second, cells must selectively be able to take up this resource, and third, the resource should be metabolized within cells to produce glucose/a glucose-like product capable of fueling a glycolytic and PPP-active state. Further, if this were indeed a 'controlling resource' that determined the emergence of light cells, preventing the uptake and utilization of this resource should prevent the emergence and proliferation of only the light cells, but leave the dark cells unaffected. In order to identify such a candidate metabolite, we considered all possible outputs of

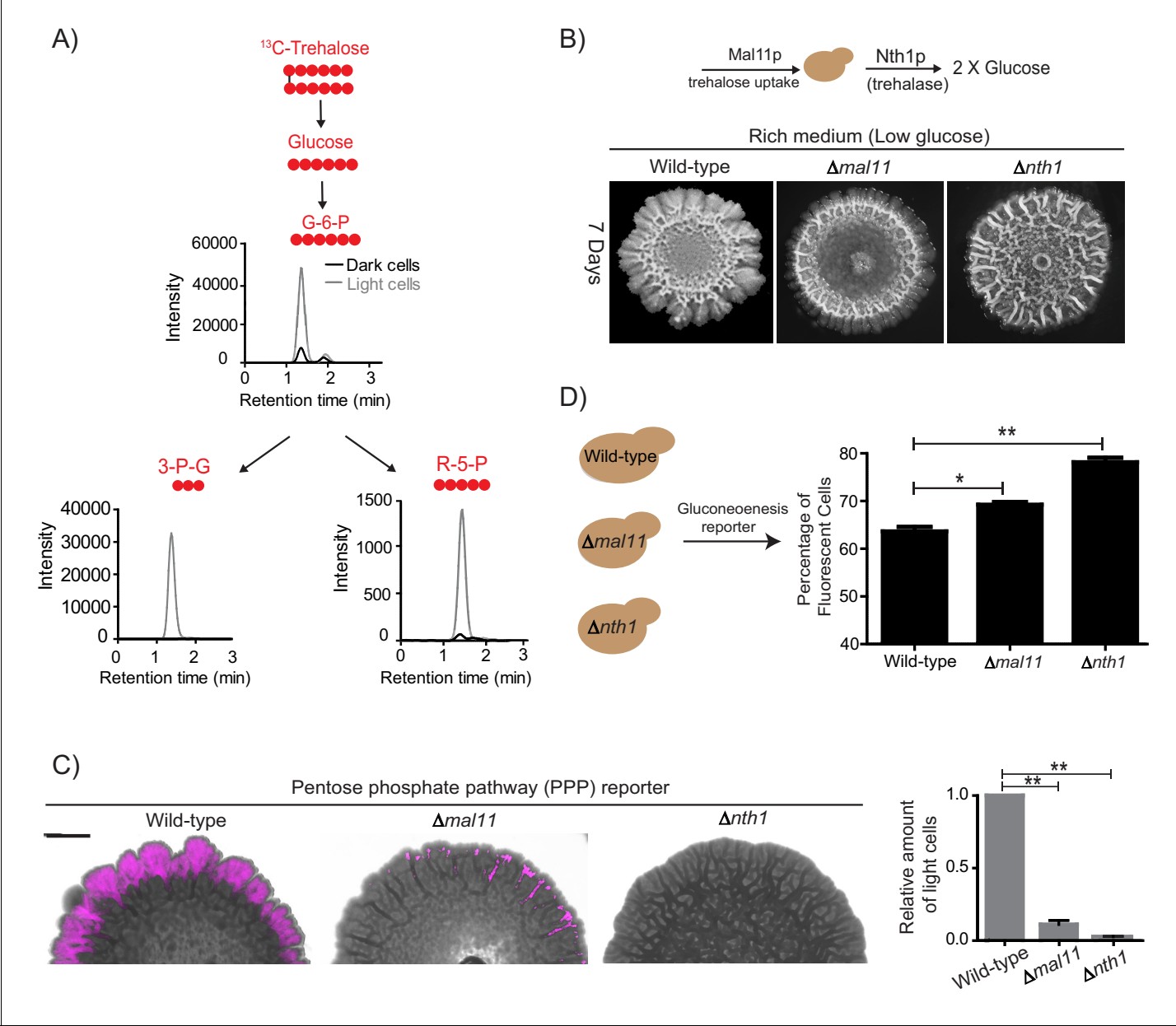

**Figure 5.** Trehalose uptake and utilization determines the existence of light cells. (A) Estimation of trehalose uptake and breakdown/utilization in light and dark cells. LC-MS/MS based metabolite analysis, using exogenously added $^{13}$C Trehalose, to compare breakdown and utilization of $^{13}$C Trehalose for glycolysis and the PPP, in light and dark cells. The red circles represent $^{13}$C labeled carbon atoms. Data for $^{13}$C labeled glycolytic and PPP intermediates (derived from trehalose) are shown. The data presented is from a single flux experiment, which was repeated independently (with different colonies) twice (n = 2). Also see *Figure 5—figure supplement 1A*. (B) Comparative development of wild-type colonies (same image used in *Figure 1A*) with colonies lacking the major trehalose transporter (Δ*mal11*), or the intracellular neutral trehalase (Δ*nth1*). Colonies are shown after 7 days of development. Scale bar: 2 mm. (C) Visualization (left panel) and quantification (right bar graphs) of light cells in wild-type (same image used in *Figure 2C*), Δ*mal11*, or Δ*nth1* cells, based on fluorescence emission dependent upon the PPP reporter activity. The quantification is based on flow cytometry data (n = 3). Scale bar: 2 mm. (D) Estimate of the percentage of gluconeogenic cells in wild-type, Δ*mal11* and Δ*nth1* (strains that cannot uptake or breakdown trehalose). This was based on quantifying the expression of the gluconeogenesis reporter plasmid (pPCK1-mCherry), expressed in all these cells. Cells from the entire colony were isolated and percentage of fluorescent cells (i.e. cells expressing the gluconeogenic reporter) in each colony was calculated by analyzing the samples by flow cytometry (n = 3). Statistical significance was calculated using unpaired t test (* indicates p<0.05, ** indicates p<0.01) and error bars represent standard deviation.

The online version of this article includes the following figure supplement(s) for figure 5:

**Figure supplement 1.** Comparative breakdown of labeled trehalose by distinct cells in a colony.

gluconeogenesis: the storage carbohydrates/sugars glycogen and trehalose, the polysaccharides of the cell wall (chitin, mannans, glycans), and glycoproteins (*Figure 4A*) (*Jules et al., 2008*; *Kayikci and Nielsen, 2015*). The large molecular size of glycogen, chitins, and complex glycosylated proteins, the lack of known cellular machinery for their uptake, and the difficulty in efficiently breaking them down make them all unlikely candidates to be the resource controlling the emergence of light cells. Contrastingly, trehalose has unique properties making it a plausible candidate. It is a small, non-reducing disaccharide composed of two glucose molecules. Trehalose has been observed in the extracellular environment in yeast (*Parrou et al., 2005*), and yeast can uptake trehalose through disaccharide transporters (*Jules et al., 2008*; *Stambuk, 1998*). Further, trehalose can be rapidly and specifically hydrolyzed to two glucose molecules, which can fuel glycolysis and re-entry into the cell division cycle (*Laporte et al., 2011*; *Shi et al., 2010*; *Shi and Tu, 2013*). These diverse data therefore presented trehalose as an excellent putative candidate metabolite that controlled the emergence of cells in the light state. To test this possibility, we first measured extracellular trehalose in colonies. Free trehalose was readily detectable in the extracellular environment of these colonies (*Figure 4B*). To test if trehalose could be differentially transported into either light or dark cells, we first estimated amounts of a primary trehalose transporter, Mal11 (*Stambuk, 1998*) in these cells. Mal11 protein amounts were substantially higher in the light cells compared to the dark cells (*Figure 4C*, and quantified in *Figure 4—figure supplement 1A*). To unambiguously, directly estimate trehalose uptake, we isolated light and dark cells from a mature colony, and exogenously added $^{13}$C-trehalose. We then measured intracellular levels of labeled trehalose present in these cells, by extracting and estimating metabolite amounts (by LC/MS/MS) (see *Table 1* for MS parameters). Notably, the light cells rapidly accumulated $^{13}$C-trehalose (*Figure 4D*), while the dark cells did not, suggesting robust, preferential uptake of extracellular trehalose.

Finally, we estimated the ability of light and dark cells to break-down and utilize trehalose. For this, we first measured the expression of the predominant neutral trehalase in yeast (Nth1) (*Jules et al., 2008*), in the light and dark cells. Light cells had substantially higher Nth1 amounts than the dark cells (*Figure 4E*, and quantified data shown in *Figure 4—figure supplement 1B*). We also measured enzymatic activity for Nth1 (in vitro, using cell lysates), and found that the light cells had ~2 fold higher in vitro enzymatic activity, compared to the dark cells (*Figure 4F*). Collectively, these data suggested that the light cells were uniquely able to preferentially take up more trehalose, break it down to glucose, in order to potentially utilize it to sustain a metabolic state with high PPP activity.

## Trehalose uptake and utilization determines the existence of light cells

Since these data suggested that trehalose uptake and utilization would be preferentially high in the light cells, we directly tested this using a quantitative metabolic flux based approach. For this we used stable-isotope labeled trehalose, and measured trehalose uptake, breakdown and utilization. To the isolated light and dark cells, $^{13}$C-labeled trehalose was externally provided, and intracellular metabolites extracted from the respective cells. The intracellular amounts of $^{13}$C -labeled glycolytic and PPP intermediates were subsequently measured using LC/MS/MS (*Figure 5A* and *Figure 5—figure supplement 1A*, also see *Table 1* for MS parameters). $^{13}$C –labeled glucose-6-phosphate (which enters both glycolysis and the PPP), the glycolytic intermediates glyceraldehyde-3-phosphate and 3-phosphoglycerate, and the PPP intermediates 6-phosphogluconate, ribulose-5-phosphate and sedoheptulose-7-phosphate all rapidly accumulated exclusively in the light cells (*Figure 5A*, *Figure 5—figure supplement 1A*). Since the labeled carbon can come only from trehalose, these data indicate both the breakdown of trehalose to glucose, as well as the subsequent utilization of glucose for these pathways. Indeed, the labeled forms of these metabolites were only above the detection limit in dark cells (*Figure 5A* and *Figure 5—figure supplement 1A*). Thus, these data demonstrate that external trehalose is preferentially taken up only by the light cells, and utilized to fuel the complimentary metabolic state of the light cells, with high glycolytic and PPP activity.

Finally, we tested if the sharing and differential utilization of trehalose determined both the emergence and the proliferation of light cells. An explicit prediction is made both in our model, and our hypothesis based on these experimental data. This is: preventing uptake and/or utilization of trehalose should prevent cells from switching to the light state. To test this prediction, we generated strains lacking *NTH1* (which cannot break-down trehalose to glucose), and *MAL11* (which will have reduced trehalose uptake), allowed them to develop into mature colonies, and compared the

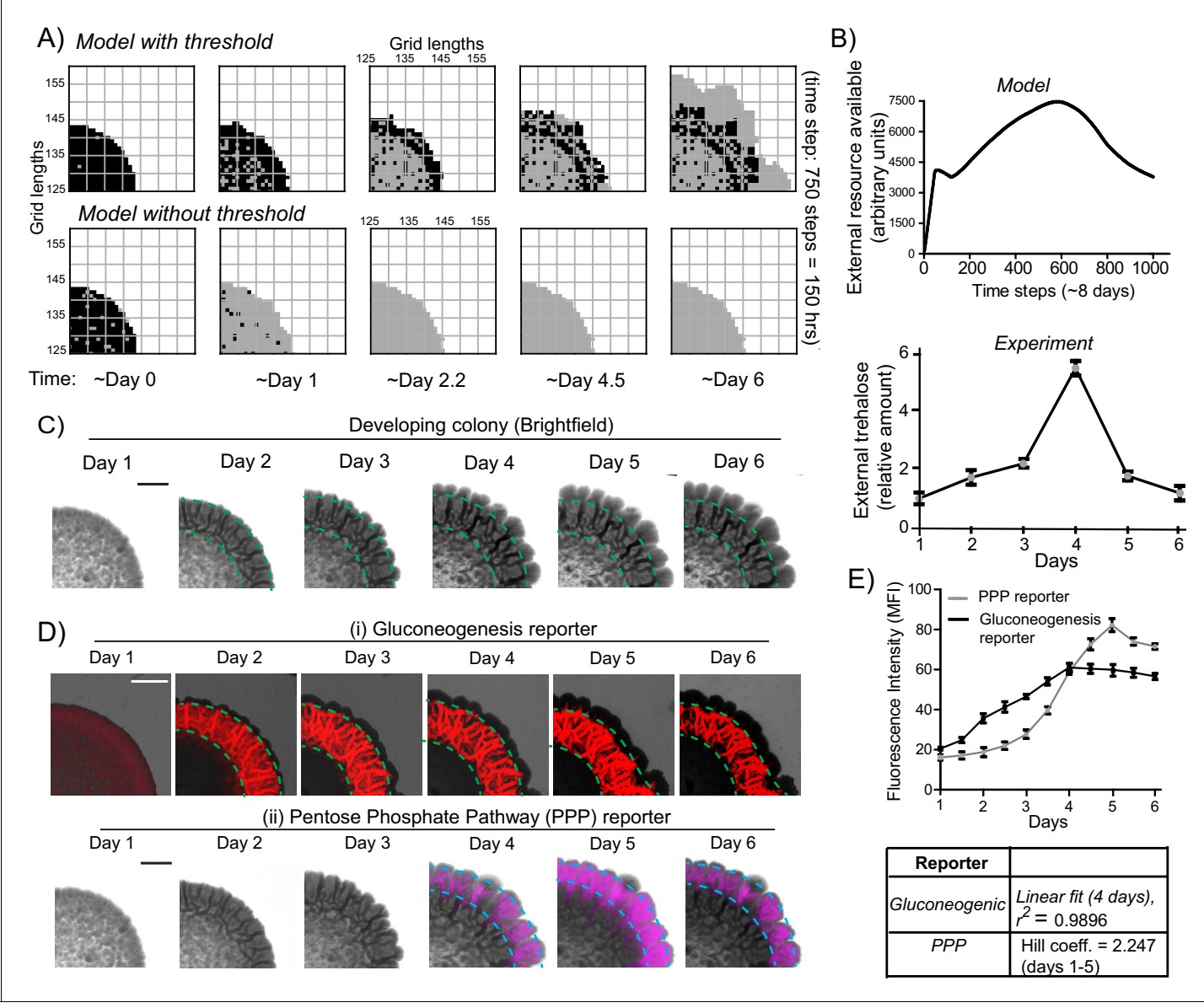

**Figure 6.** A resource threshold effect controls cooperative switching of cells to the light state. (A) Simulation of colony development, based on the default model (which incorporates a resource threshold buildup, followed by consumption, switching to a light state, and expansion), compared to a model where the threshold amounts of the resource is removed. Note the final expansion size of the colony. Also see *Figure 6—figure supplement 1A–D* and *Video 5*. (B) (i) Changes in the availability of the resource as the colony develops, based on the model. (Ii) Extracellular amounts of trehalose measured from developing wild-type colonies. Data from three independent colonies. Note: in the model, in ~3–4 days the resource is highest, and reduces sharply after that. In the experimentally obtained data, extracellular trehalose amounts are highest at ~day 4, and then rapidly decreases over day 5. This correlates to when the light cells emerge and expand. (C) A time-course of bright-field images of the developing wild-type colony, illustrating the distribution of dark cells, and the emergence and distribution of light cells. (D) A time-course revealing fluorescence based estimation of the (i) reporter for gluconeogenic activity (dark cells), or (ii) the PPP activity reporter (light cells). Note the delayed, rapid appearance and increase in the PPP activity reporter. (E) Quantification of the increase in the gluconeogenic reporter activity in the colony, and the PPP reporter activity (based on fluorescence intensity) within the colony. The increase in gluconeogenic reporter activity, when plotted, is linear, and saturates. The increase in PPP activity over the first 5 days is highly cooperative (as estimated using a Hill coefficient as a proxy for cooperativity), before saturating (n = 3). Error bars represent standard deviation.

The online version of this article includes the following figure supplement(s) for figure 6:

**Figure supplement 1.** Colonies generated using different switching rules.

amounts of light cells in each. Compared to wild-type colonies, cells lacking the major trehalose

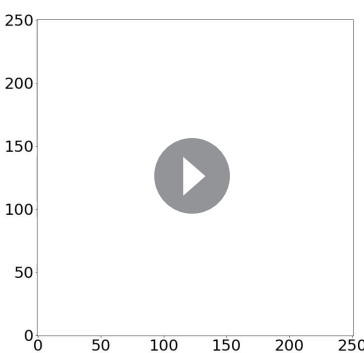

**Video 5.** Development of colony without a resource threshold. Simulation video showing changes in a colony where the dark cells switch to light by random chance (probability p=0.5). They don't need the resource levels to reach a certain threshold. Once they become light cells, they cannot switch back to dark. Due to lack of the produced resource, the colony doesn't grow discernibly.

https://elifesciences.org/articles/46735#video5

uptake transporter (Δmal11) formed colonies with very few light cells (*Figure 5B*). Note: while Mal11 shows a high affinity for trehalose, *S. cerevisiae* has multiple sugar transporters with reduced affinity for any disaccharide. Therefore, cells lacking *MAL11* may take up trehalose with lower efficiency. In these cells, the ability to break-down trehalose remains intact. More importantly, in colonies of cells lacking trehalase (Δnth1), and which therefore cannot efficiently breakdown internal trehalose, had nearly no detectable light cells, based on brightfield microscope observations (*Figure 5B*). This result was more quantitatively estimated in colonies of cells with these respective genetic backgrounds, using the expression of the fluorescent PPP reporter. Again, almost no PPP reporter activity was observed in the Δnth1 cell colonies, while very few cells with PPP reporter activity were seen in Δmal11 colonies (*Figure 5C*). As controls, we ensured that there were no defects in the expression of the PPP reporters in cells from these genetic backgrounds. Correspondingly, we also quantified the percentage of dark, highly gluconeogenic cells (as determined using the gluconeogenesis reporter), in colonies from each of these genetic backgrounds. The percentage of gluconeogenic cells was proportionately higher in the Δmal11 (~73%), and Δnth1 (~80%) colonies compared to the wild-type colony (~65%) (*Figure 5D*). Thus, controlling the uptake and utilization of the resource (trehalose) directly regulates the emergence of cells in the light state.

Collectively, these data demonstrate that trehalose is the shared gluconeogenic resource that determines the emergence, and persistence, of light cells within the structured colony.

## A resource threshold effect controls cooperative switching of cells to the light state

Our experimental data showing the organization of dark and light cells was obtained from ~5–6 day old, mature colonies. However, in our simulations of the temporal development of the colony, we observed that the dense network of dark (gluconeogenic) cells form first, followed by a very late appearance of light cells (*Figure 6A* and *Video 1*). This late appearance of light cells in the simulations comes from an inherent threshold effect included within the model. Here, the external build-up of the shared resource made by the dark cells is required. At a sufficient built-up concentration, this resource will trigger the switching of some cells to light cells. Light cells in turn will consume the resource, reducing the available amounts, thereby preventing other cells from switching to this new state. This threshold-effect therefore predicts a delayed, rapid emergence of light cells, and also enables such a pattern of distinct cell groups to form. If this threshold requirement is removed in the simulation (for example when replaced by a rate of switching from dark to light that depends linearly on the amount of resource), the resultant colony remains small, and the organized pattern of cells in two states does not occur. This is shown in *Figure 6A*, and *Video 5*. This small colony size is largely due to low resource amounts to support the proliferation of the light cells, since there are insufficient dark cells remaining to produce the resource. This is also clearly seen in control simulations with a range of resource amounts, and linear switching, as shown in *Figure 6—figure supplement 1A–1D*. Contrastingly, in the model that successfully simulates the colony development, the externally available amount of the resource builds-up, reaches the threshold (where cells switch to the light state), and then rapidly decreases, if the light cells also consume the resource (*Figure 6B*, upper panel).

This therefore prompted us to more closely examine the development of actual colonies over time, for these properties. We first estimated the relative amounts of extracellular, free trehalose in

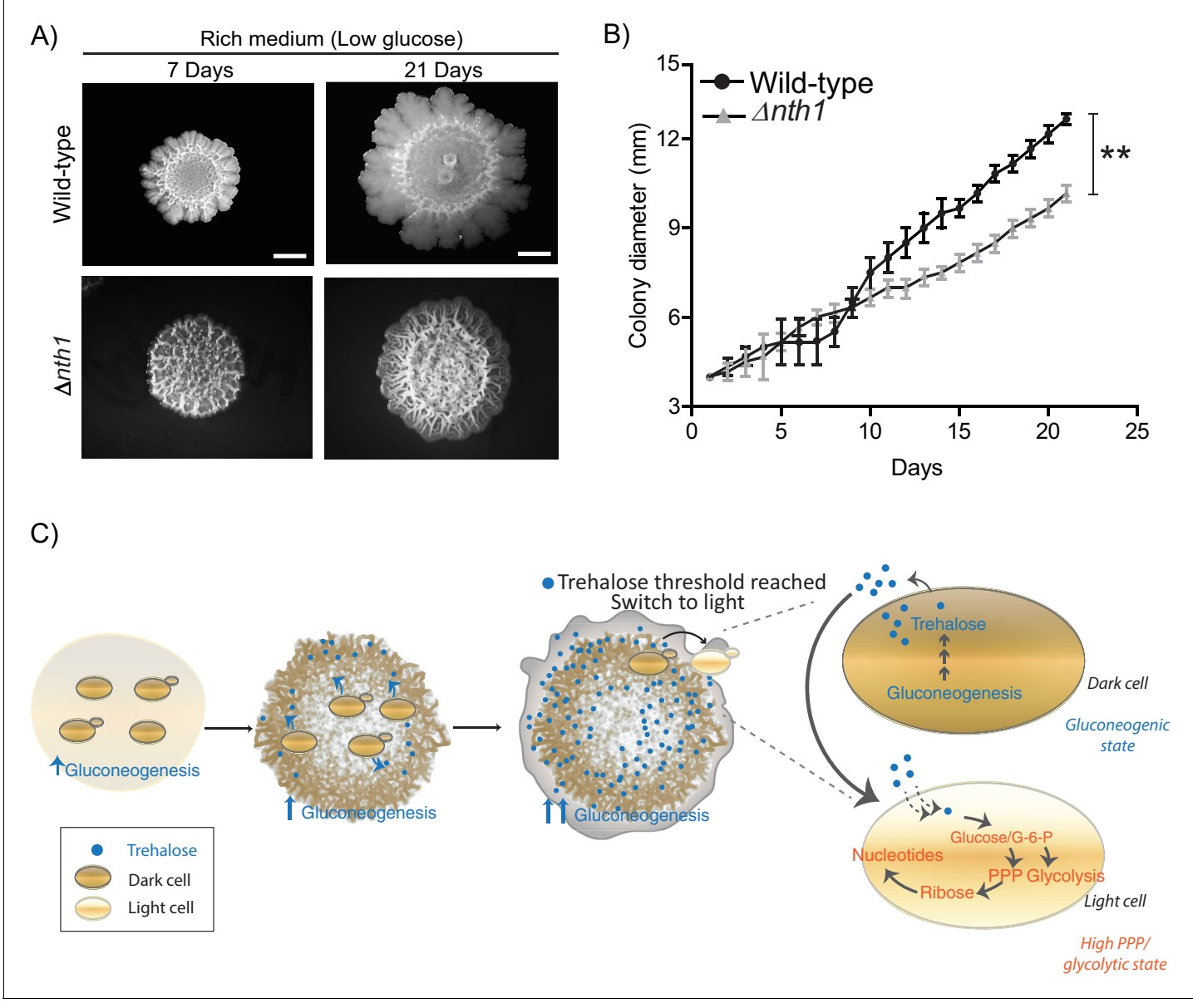

**Figure 7.** Physiological advantages of cells with organized spatial heterogeneity. (**A**) Foraging response of wild-type cells (same image used in *Figure 1A*) and Δ*nth1* cells measured as a function of their ability to spread on a plate. Colony spreading was quantified by measuring the diameter of the colonies every day for 21 days (n = 3). (**B**) Cells in low glucose perform gluconeogenesis, as required in low glucose medium. As gluconeogenic reserves build up, trehalose builds up in the extracellular environment. At a threshold concentration of trehalose, some cells switch to a high glycolytic, PPP state. This state depends upon the utilization of trehalose to fuel it. This utilization of trehalose by the light cells results in decreased external trehalose to below a threshold. This in turn restrains the other, remaining cells in a gluconeogenic state, where they continue to produce trehalose. This gives rise to the final, self-organized community, with specialization of function and division of labor. Statistical significance was calculated using unpaired t test (** indicates $p < 0.01$) and error bars represent standard deviation.

the colony over time. Notably, trehalose amounts steadily increased over 4 days, and subsequently rapidly decreased (*Figure 6B* lower panel). This rapid decrease in trehalose after day four is despite a steady, continuing increase in the total number of cells in the colony (*Figure 6—figure supplement 1E*). We next monitored the development of colonies over time, to determine when the light cells emerge. Using just the bright-field image reconstruction of the colonies, during this time course, the intensity of dark cells steadily increased, and organized into the mesh-like network over 4 days (*Figure 6C*). However, the light cells appeared only after ~4 days, and rapidly increased in number (*Figure 6C*). We more quantitatively estimated this, using strains expressing the

gluconeogenic- or the PPP-reporter (*Figure 6D*). Notably, the increase in total fluorescence intensity due to the gluconeogenic-reporter in the colony (over time) was relatively linear over the first four days ($r^2$ = 0.99), increasing with the steady increase in the number of cells (*Figure 6—figure supplement 1E*). Contrastingly, the increase in the PPP reporter activity over the first five days was clearly non-linear and switch like, with very low signal intensity for the first three days, and then a rapid emergence of signal over days four and five (*Figure 6E*). This indicated a cooperative, switch like emergence of, and increase in these light cells. A useful biophysical measure of cooperativity (more commonly used for protein-ligand binding characteristics) is the Hill coefficient. We adopted the Hill equation, using the amount of PPP reporter fluorescence (instead of ligand-receptor binding), to estimate cooperativity in the system. Over the first five days the increase in PPP-reporter activity showed a Hill coefficient greater than 1, indicating a positively cooperative switch of cells to the light state (*Figure 6E*). This nicely correlates with the build-up, and rapid decrease in external trehalose around day 4 (*Figure 6B*). These data also show that the peripheral location of the light cells cannot simply be due to possible greater access to glucose in the medium, since for the first ~4 days, there are no cells in the periphery with high PPP reporter activity. Their emergence is indeed rapid, and switch-like.

In summary, data from model simulations and experiments show that initially the gluconeogenic cells increase in number, leading to release and build-up of the resource (trehalose) in the local environment. At this time there are no light cells in the colony. Above a threshold concentration of trehalose, some cells rapidly switch to light state with high PPP activity. The further expansion of these light cells correlates with rapid consumption of the extracellular trehalose that sustains this state. These data suggest a threshold effect, where the controlling resource, trehalose, needs to build up above a certain amount, in order for cells to switch to the contrary, high PPP activity state.

## Cells in distinct metabolic states provide a collective growth advantage to the colony

Finally, we wondered if such an emergence of light cells with high PPP activity might benefit the community of cells as a whole. To address this, we compared the long term colony expansion of wild-type cells, with colonies comprised of cells lacking the neutral trehalase (Δ*nth1*). Cells in the Δ*nth1* colonies cannot utilize trehalose to produce glucose, and as shown earlier, will remain in a gluconeogenic state. Therefore, in these Δ*nth1* cell colonies light cells will be absent. However, these cells are still capable of normal gluconeogenesis (and trehalose production). Strikingly, we observed that as the respective colonies expanded over time (~21 days), the wild-type colonies spread over a significantly greater area on the plate, while the Δ*nth1* colonies were unable to expand as efficiently (*Figure 7A* and *Figure 7B*). This shows that the emergence and proliferation of light cells are important for the expansion of the colony. Since the dark cells are required for the emergence and existence of the light cells, collectively, these data suggest how the community uses cells in distinct metabolic states to maximize growth and spatial expansion, possibly to forage for new nutrients.

## Discussion

Collectively, we present a simple model proposing how cells in metabolically distinct states spontaneously emerge and spatially self-organize within a yeast colony, as summarized in *Figure 7C*. In low glucose conditions, cells begin in a uniform gluconeogenic state, which is the expected metabolic state in this nutrient condition. The gluconeogenic cells produce a resource (trehalose), that is now externally available. This resource builds up to above a threshold amount. At this threshold, some cells take up and consume trehalose, breaking it down to glucose. These cells spontaneously switch to the complimentary metabolic state, with high PPP and glycolytic activity (i.e. the light state) (*Figure 7C*). These light cells can remain in this metabolic state only so as long as the resource (trehalose) is externally available. However, as trehalose is consumed by these cells, the available amount of external trehalose itself drops below the threshold. The surrounding dark cells therefore remain trapped in a gluconeogenic state, continuing to produce trehalose. Thereby, a predictable fraction of cells, constrained spatially, will remain in each metabolic state, resulting in specialized cell groups and division of metabolic labor. Thus, biochemically heterogeneous cell states can spontaneously emerge and spatially self-organize. An implicit concept emerging from this study is that of threshold amounts of a controlling or sentinel metabolite that regulates a switch to a new metabolic

state. By definition, such a metabolite must be produced by cells present in a certain (original) metabolic state. But when this metabolite is utilized, it must have the ability to switch cells to an entirely distinct metabolic state. Further, the emergence and expansion of cells in the new state will be rapid and switch-like, resembling a bistable system (*Pomerening, 2008*). This idea of metabolites controlling cell states is an emerging area of interest (*Cai and Tu, 2011*; *Krishna and Laxman, 2018*), but has not been studied in the context of groups of cells organizing into distinct groups or metabolic states (and therefore different phenotypic properties).

We speculate what the advantages of such spatially organized, phenotypically distinct states within a group of clonal cells might be, in this example. Here, we observe that the organized community with cells in distinct states has clear advantages, in being able to spatially expand better (*Figure 7C*). For sessile microbes such as yeast, this ability to forage for better nutrients is important for their survival. This might also convey other advantages, and uncovering those are obvious areas of future studies. Since the inherent properties of the cells in the distinct states are different, this raises the deeper possibility that these advantages come from physical and chemical properties of the cells, which arise from their distinct metabolic states. Regardless, our study substantially advances descriptions of yeast 'multicellularity' from simple dimorphism, aggregated cells, or three-dimensional colony forms (*Cáp et al., 2012*; *Koschwanez et al., 2011*; *Palková and Váchová, 2016*; *Ratcliff et al., 2012*; *Váchová and Palková, 2018*; *Veelders et al., 2010*; *Wloch-Salamon et al., 2017*), to self-organized, phenotypically heterogeneous cell states exhibiting division of labor and metabolic interdependence. Strikingly, the nature of spatial patterning allied with division of labor that we observe in yeast is reminiscent of true multicellular systems (*Newman, 2016*; *Niklas, 2014*). Also, the cell states in these yeast colonies can be considered commensal, since trehalose is a necessary output of gluconeogenesis, and therefore a default, biochemically non-limiting output in dark cells. Since trehalose controls the emergence and maintenance of light cells in the complimentary metabolic state, it thus can be considered a resource benefiting the light state. Thus simple, metabolism-derived constraints are sufficient to determine how contrary biochemical states can spontaneously emerge and be supported, in conjunction with spatial structure. Such organization of cells into specialized, labor-divided communities expands on the role of reaction-diffusion systems (particularly activator-depleted substrate schemes) in controlling cellular patterning (*Gierer and Meinhardt, 1972*; *Kondo and Miura, 2010*; *Newman, 2016*), with a metabolic resource threshold being central to the emergence and stabilization of a new phenotype (*Cai and Tu, 2011*; *Krishna and Laxman, 2018*). A deeper dissection of what such constraints can permit will therefore advance our general understanding of how specialized cell states can emerge and be stabilized.

Metabolic cross-feeding is best understood currently in multi-species microbial communities, where this has been inferred largely using inter-species genomic comparisons (*Ackermann, 2015*; *D'Souza et al., 2018*; *Goldford et al., 2018*; *Tyson et al., 2004*). Further, metabolic sharing has typically been demonstrated using synthetically engineered systems where mutual dependencies are created (*Campbell et al., 2016*; *D'Souza et al., 2018*; *Mee et al., 2014*; *Pande et al., 2015*; *Wintermute and Silver, 2010*). The spatial organizations of any such populations remain challenging to model. Biochemically identifying metabolites that are conclusively exchanged between cooperating cells remains difficult, and the significance of such putative metabolite exchange challenging to interpret (*Ackermann, 2015*; *D'Souza et al., 2018*). Finally, such studies have emphasized non-isogenic systems, where genetic changes stabilize different phenotypes, and auxotrophies define the nutrient sharing or cooperation (*Ackermann, 2015*). Contrastingly, here we directly identify a produced metabolic resource, and demonstrate how its availability and differential utilization can control the emergence of cells in opposing metabolic states, in a clonal population. We also explain how the spontaneous spatial organization into phenotypically distinct cell groups occurs. Thus, our study also goes beyond stochastic gene expression (*Ackermann, 2015*; *Balázsi et al., 2011*; *Blake et al., 2003*) to explain how phenotypic heterogeneity and specialization can emerge in clonal populations. By considering these metabolism-derived rules, and thereby manipulating available metabolic resources, we suggest how it can be viable to program the formation, structure or phenotypic composition of isogenic cell populations. Collectively, such simple physico-chemical constraints can advance our understanding of how isogenic cells can self-organize into specialized, labor-divided groups, as a first step towards multicellularity.

## Materials and methods

### Yeast strains and growth media

The prototrophic sigma 1278b strain (referred to as wild-type or WT) was used in all experiments. Strains with gene deletions or chromosomally tagged proteins (at the C-terminus) were generated as described (*Longtine et al., 1998*). Strains used in this study are listed in *Table 2*. The growth medium used in this study is rich medium (1% yeast extract, 2% peptone and 2% glucose or 0.1% glucose).

### Colony spotting assay

All strains were grown overnight at 30 ˚C in either rich medium or minimal medium. 5 microliters of the overnight cultures were spotted on rich medium (low glucose) (1% yeast extract, 2% peptone, 0.1% glucose and 2% agar). Plates were incubated at 30 ˚C for 7 days unless mentioned otherwise.

### Colony imaging

For observing colony morphology, colonies were imaged using SZX-16 stereo microscope (Olympus) wherein the light source was above the colony. Bright-field imaging of 7 day old colonies were done using SZX-16 stereo microscope (Olympus) wherein the light source was below the colony. Epifluorescence microscopy imaging of 7 day old gluconeogenesis reporter colonies (pPCK1-mCherry), pentose phosphate pathway (PPP) reporter colonies (pTKL1-mCherry) and *HXK1* reporter colonies (pHXK1-mCherry) were imaged using the red filter (excitation of 587 nm, emission of 610 nm) of SZX-16 stereo microscope (Olympus). Similar protocol was followed for imaging 1 day to 6 day old colonies.

### Analysis of fluorescent cell populations in reporter strain colonies

Light cells and dark cells isolated from 7 day old wild-type colonies harboring either the gluconeogenesis reporter, PPP reporter or the *HXK1* reporter were re-suspended in 1 ml of water. The percentage of fluorescent cells were determined by running the samples through a flow cytometer, and counting the total number of mCherry positive cells in a total of 1 million cells. Light cells and dark cells isolated from wild-type colonies without the fluorescent reporter were used as control.

### Biochemical estimation of trehalose/glycogen levels

Trehalose and glycogen from yeast samples were quantified as described previously, with minor modifications (*Shi et al., 2010*). 10 $OD_{600}$ of light cells and dark cells from 7 day old wild-type colonies (rich medium, 0.1% glucose) were collected. After re-suspension in water, 0.5 ml of cell suspension was transferred to four tubes (two tubes for glycogen assay and the other two tubes for trehalose assay). When sample collections were complete, cell samples (in 0.25 M sodium carbonate) were boiled at 95–98˚C for 4 hr, and then 0.15 ml of 1 M acetic acid and 0.6 ml of 0.2 M sodium acetate were added into each sample. Each sample was incubated overnight with 1 U/ml amyloglucosidase (Sigma-Aldrich) rotating at 57˚C for the glycogen assay, or 0.025 U/ml trehalase (Sigma-Aldrich) at 37˚C for the trehalose assay. Samples were then assayed for glucose using a glucose assay kit (Sigma-Aldrich). Glucose assays were done using a 96-well plate format. Samples were added into each well with appropriate dilution within the dynamic range of the assay (20–80 µg/ml glucose). The total volume of sample (with or without dilution) in each well was 40 microliters. The plate was pre-incubated at 37˚C for 5 min, and then 80 µl of the assay reagent from the kit was added into each well to start the colorimetric reaction. After 30 min of incubation at 37˚C, 80 microliters of 12 N sulfuric acid was added to stop the reaction. Absorbance at 540 nm was determined to assess the quantity of glucose liberated from either glycogen or trehalose. For measurement of extracellular trehalose measurement, single wild-type colony (1 day to 7 day old colony) was re-suspended in 100 microliters of water and centrifuged at 20000 g for 5 min. Supernatant was collected and buffered to a pH of 5.4 (optimal for trehalase activity) using sodium acetate buffer (pH 5.0). 0.025 U/ml trehalase (Sigma-Aldrich) was added and samples were incubated at 37˚C overnight. Glucose concentration was estimated as described earlier.

## Neutral trehalase activity assay

Neutral trehalase activity assay was performed as described earlier with the following modifications (*De Virgilio et al., 1991*). Briefly, 1 $OD_{600}$ of light cells and dark cells isolated from 7 day old wild-type colonies (rich medium, 0.1% glucose) were washed twice with ice-cold water. For permeabilization, cells were re-suspended in tubes containing equal volume of 1% Triton-X in assay buffer (200 mM tricine buffer ($Na^+$) (pH 7.0)) and immediately frozen in liquid nitrogen. After thawing (1–4 min at 30°C), the cells were centrifuged (2 min at 12000 g), washed twice with 1 ml of ice-cold assay buffer and immediately used for the trehalase assay. Trehalase assay was performed in 50 mM tricine buffer ($Na^+$) (pH 7.0), 0.1 M trehalose, 2 mM manganese chloride ($MnCl_2$) and the Triton X-100 permeabilized cells in a total volume of 400 microliters. After incubation for 30 min at 30°C, the reaction was stopped in a boiling water bath for 3 min. Glucose concentration in the supernatant was determined using the glucose assay kit (Sigma-Aldrich).

## Western blot analysis

Approximately ten $OD_{600}$ cells were collected from respective cultures, pelleted and flash frozen in liquid nitrogen until further use. The cells were re-suspended in 400 microliters of 10% trichloroacetic acid (TCA) and lysed by bead-beating three times: 30 s of beating and then 1 min of cooling on ice. The precipitates were collected by centrifugation, re-suspended in 400 microliters of SDS-glycerol buffer (7.3% SDS, 29.1% glycerol and 83.3 mM tris base) and heated at 100°C for 10 min. The supernatant after centrifugation was treated as the crude extract. Protein concentrations from extracts were estimated using bicinchoninic acid assay (Thermo Scientific). Equal amounts of samples were resolved on 4% to 12% bis-tris gels (Invitrogen). Western blots were developed using the antibodies against the respective tags. We used the following primary antibody: 538 monoclonal FLAG M2 (Sigma-Aldrich). Horseradish peroxidase–conjugated secondary antibody (anti-mouse) was obtained from Sigma-Aldrich. For Western blotting, standard enhanced chemiluminescence reagents (GE Healthcare) were used.

## Metabolite extractions and measurements by LC-MS/MS

Light cells and dark cells isolated from wild-type colonies grown in different media were rapidly harvested and metabolites were extracted as described earlier (*Walvekar et al., 2018*). Metabolites were measured using LC-MS/MS method as described earlier (*Walvekar et al., 2018*). Standards were used for developing multiple reaction monitoring (MRM) methods on Sciex QTRAP 6500. Metabolites were separated using a Synergi 4μ Fusion-RP 80A column (100 × 4.6 mm, Phenomenex) on Agilent's 1290 infinity series UHPLC system coupled to the mass spectrometer. For positive polarity mode, buffers used for separation were- buffer A: 99.9% $H_2O$/0.1% formic acid and buffer B: 99.9% methanol/0.1% formic acid (Column temperature, 40°C; Flow rate, 0.4 ml/min; T = 0 min, 0% B; T = 3 min, 5% B; T = 10 min, 60% B; T = 11 min, 95% B; T = 14 min, 95% B; T = 15 min, 5% B; T = 16 min, 0% B; T = 21 min, stop). For negative polarity mode, buffers used for separation were- buffer A: 5 mM ammonium acetate in $H_2O$ and buffer B: 100% acetonitrile (Column temperature, 25°C; Flow rate: 0.4 ml/min; T = 0 min, 0% B; T = 3 min, 5% B; T = 10 min, 60% B; T = 11 min, 95% B; T = 14 min, 95% B; T = 15 min, 5% B; T = 16 min, 0% B; T = 21 min, stop). The area under each peak was calculated using AB SCIEX MultiQuant software 3.0.1.

## $^{15}$N- and $^{13}$C- based metabolite labeling experiments

For detecting $^{15}$N label incorporation in nucleotides, $^{15}$N Ammonium sulfate (Sigma-Aldrich) and $^{15}$N Aspartate (Cambridge Isotope Laboratories) with all nitrogens labeled were used. For $^{13}$C-labeling experiment, $^{13}$C Trehalose with all carbons labeled (Cambridge Isotope Laboratories) was used. All the parent/product masses measured are enlisted in *Table 1*. For all the nucleotide measurements, release of the nitrogen base was monitored in positive polarity mode. For all sugar phosphates, the phosphate release was monitored in negative polarity mode. The HPLC and MS/MS protocol was similar to those explained above.

## Building and implementing the model

The model simulation code is available via GitHub ref: https://github.com/vaibhhav/yeastmetabcolony.

## Components

The model consists of (i) a population of light and dark cells, and (ii) a shared metabolic resource that is produced by, and is accessible to the cells. Therefore, the dynamic processes involved can be broadly divided into those pertaining to the cells of the colony and those pertaining to the shared resource. The cells and resource occupy a 2-D square grid, which represents the surface of an agar plate. If one takes each grid length to correspond to 50 µm in real space, then, given the average size of a yeast cell at 5 µm, a single grid location can be imagined to contain upto 100 cells, which we term 'cell blocks'. We coarse-grain the model such that each location is either empty, occupied by light cell block, or a dark cell block. That is, we ignore the possibility that cell blocks might be mixed. This is simply for computational ease. A more detailed model consisting of smaller grid lengths such that each location could hold at most a single cell would exhibit the same behavior as the coarse-grained one, but would require much larger grid sizes and longer computational times in order to simulate realistic sized colonies. With the coarse-graining, our simulations use a 250 × 250 grid. Each grid location also contains saturating amounts of amino acids, as well as a certain level of the shared metabolic resource. If a location has a cell block, that block also has internal levels of the amino acids and the resource, which may be different from the external level in that location.

## Initial state of the grid

We start with an approximately circular colony 20 grid lengths in radius (covering 1257 grid locations) in the center of the 250 × 250 grid. 95–99% of these 1257 cell blocks are in the dark state, while 1–5% are in the light state, distributed randomly in the colony. The concentration or level of the shared resource is set to zero at every location. However, at all times, we assume the presence, throughout the grid, of saturating amounts of amino acids that are required for the (slow) growth of the dark cells.

## Dynamics of the model

The grid is updated at discrete time steps. Each time step corresponds to 12 min in real time, and all simulations are run for 750 time steps, that is 150 hours of real time (~6 days). In each time step, we first go over every cell block to implement the following processes:

If a block at location (x,y) is dark, then:

1. If the resource level at (x,y) is above a certain threshold $S$ = 3.0 units of resource, then the cells in the block switch to being light cells with a probability $p$=0.5
2. If the block is still dark, then add $R$ = 0.07 units to the resource level at (x,y).
3. Consume (internalize) $C$ = 0.05 units of amino acids (present in saturating amounts at all locations)
4. If the internal amino acid level is greater than or equal to 1.0, the dark block can divide with a probability $g_d$ = 0.01.
5. If the block can divide, then check if there's an empty location in the immediate neighborhood. The immediate neighborhood is the set of locations {(x-1,y), (x + 1,y), (x,y-1), (x,y + 1)}.
6. If there's at least one empty space, preferably divide into an empty location which has more occupied neighbors. After division, the two daughter blocks are each assigned half the internal amino acid reserves of the original mother block.

If a block at location (x,y) is light, then:

1. If the resource level at (x,y) is greater than or equal to $C$ = 0.05 units, consume (internalize) all of it.
2. If the internal resource level is greater than or equal to 1.0, the dark block can divide with a probability $g_l$ = 0.04.
3. If the block can divide, then check if there's an empty location in the immediate neighborhood. The immediate neighborhood is the set of locations {(x-1,y),(x + 1,y),(x,y-1), (x,y + 1)}.
4. If there's at least one empty space, preferably divide into an empty location that has more occupied neighbors. After division, the two daughter blocks are each assigned half the internal resource reserves of the original mother block.

(The above set of rules and parameters is for simulating the wild-type colony. For the variations highlighted in the main text (*Figures 3E* and *6A*, bottom row), see the 'Variants of the wild-type model' section below.)

These processes implement growth of cells, as well as production and consumption of amino acids and the shared metabolic resource. Subsequent to this, in each time step, we allow diffusion of the resource levels across the grid (the "external" level at the location, not the internal levels in cell blocks), using a numerical scheme called Forward Time Central Space (FTCS). Say that the value of the resource at time t and location (x,y) is given by $U^t_{x,y}$. The FTCS scheme updates the value simultaneously at all locations using the following formula:

$$U^{t+\Delta T}_{x,y} = U^t_{x,y} + D\frac{\Delta T}{\Delta L^2}\left( U^t_{x-1,y} +\ U^t_{x+1,y} +\ U^t_{x,y-1} +\ U^t_{x,y+1} -\ 4U^t_{x,y}\right)$$

where $\Delta T$ is the time step and $\Delta L$ is the space step, or grid length, and $D$ is the diffusion constant for the resource.

## Model parameters

1. The parameters of the model are shown in *Table 3*. Time and length units are chosen such that each time step is one unit of time, and each grid length is one unit length. With these choice of units, the growth parameters for light and dark cells, respectively, are $g_l$ = 0.04, $g_d$ = 0.01. These were chosen so as to reflect the relative rates of diffusion and division. Light cells were observed to grow faster than the dark cells, so their respective growth parameters are set accordingly.
2. The switching threshold parameter (S = 3.0) was chosen to account for a delay in the switching of dark cells to another metabolic pathway via nutrient sensing as well as to give a reproducible facsimile of the experimental colonies.
3. The shared resource production value was chosen to be 7% (R = 0.07) of the minimum required to divide. In each time step, every block of dark cells adds this amount to the resource grid. This was chosen as a default value, which gave a reproducible facsimile of the experimental colonies. Other values were tried and their effect is seen in *Figure 3—figure supplement 1C*.
4. All cells consumed a small level of metabolites (the shared resource or amino acids) in each time step. This value was chosen to be 5% of the minimum required for division (C = 0.05). This gave division times that approximately matched the division times observed experimentally.
5. The switching probability (p=0.5) was chosen to add an element of stochasticity. So even if the threshold resource conditions are met, dark cells have a 50% chance to switch to light cells in that time step.
6. The choice of the diffusion constant (D = 0.24) is limited by the numerical stability of the FTCS scheme, which allows only a maximum value of D = 0.25. In real time and length units, this corresponds to a diffusion constant $D_{eff}$ of 8.7 × $10^{-13}$ m$^2$/s. $D_{eff}$ is an order of magnitude smaller than the diffusion constant for sugars like glucose and sucrose in water (*Roache, 1972*). Since the agar used for the experiments is mostly water, the diffusion constants in water can be considered as a good reference point.

## Variants of the wild-type model in different figures

*Figure 3E*, *Figure 6A* (bottom row) and *Figure 6—figure supplement 1C–D* showcase some of the final colonies generated by the simulations when the rules described above are varied. The following changes were made to the rules/parameters to generate these. 3E(i): No sharing: Set R = 0.

3E(ii): No switching from dark to light state: Set p=0.

3E(iii): 'Reverse' sharing: Set R = 0. When a cell block is light it adds R'=0.07 to the amino acid grid at the same location.

6A(bottom row): No resource thresholding: Set S = 0.

*Figure 6—figure supplement 1C–D*: Linear switching: Set S = 0. The probability of switching from dark to light state, *p*, is now a linear function of the locally available resource with a maximum value of 1.0. That is, $p = \max (m * U^t_{x,y}, 1.0)$, where *m* is a parameter that sets the slope of this linear relationship.

## Acknowledgements

This work was supported by a Wellcome Trust-DBT India Alliance Intermediate Fellowship (IA/I/14/2/501523) and institutional support from inStem and the Department of Biotechnology (DBT), Govt. of India to SL, a Wellcome Trust-DBT India Alliance Early Career Fellowship (IA/E/16/1/502996) to SV, institutional support from NCBS-TIFR and the Simons Foundation to VS and SK. The authors thank Vidyanand Nanjundiah, Ramray Bhat and Andras Paldi for insightful comments on this manuscript. We acknowledge the extensive use of the NCBS/inStem/CCAMP mass spectrometry and Central Imaging and Flow facilities.

## Additional information

### Funding

| Funder | Grant reference number | Author |
| --- | --- | --- |
| Wellcome Trust/DBT India Alliance | IA/I/14/2/501523 | Sunil Laxman |
| Simons Foundation | | Vaibhhav Sinha<br>Sandeep Krishna |
| Wellcome Trust/DBT India Alliance | IA/E/16/1/502996 | Sriram Varahan |

The funders had no role in study design, data collection and interpretation, or the decision to submit the work for publication.

### Author contributions

Sriram Varahan, Conceptualization, Data curation, Formal analysis, Funding acquisition, Validation, Investigation, Visualization, Writing—original draft; Adhish Walvekar, Data curation, Formal analysis, Validation, Investigation, Methodology, Writing—review and editing; Vaibhhav Sinha, Resources, Data curation, Software, Validation, Investigation, Visualization, Methodology, Writing—original draft; Sandeep Krishna, Data curation, Software, Formal analysis, Supervision, Funding acquisition, Methodology, Writing—original draft, Writing—review and editing; Sunil Laxman, Conceptualization, Resources, Data curation, Formal analysis, Supervision, Funding acquisition, Validation, Investigation, Visualization, Methodology, Writing—original draft, Project administration, Writing—review and editing

### Author ORCIDs

Sriram Varahan ORCID https://orcid.org/0000-0002-3609-4032
Adhish Walvekar ORCID https://orcid.org/0000-0001-7344-7653
Vaibhhav Sinha ORCID https://orcid.org/0000-0002-5169-5485
Sandeep Krishna ORCID https://orcid.org/0000-0002-0581-173X
Sunil Laxman ORCID https://orcid.org/0000-0002-0861-5080

### Decision letter and Author response

Decision letter https://doi.org/10.7554/eLife.46735.sa1
Author response https://doi.org/10.7554/eLife.46735.sa2

## Additional files

### Supplementary files

- Source code 1. Simulation codes in Python.
- Transparent reporting form

## Data availability

Mathematical model data in this study are generated by computational simulations. All model parameters and equations are included in the text, and source code is included with this article.

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
