## [Decision Letter]

Thank you for submitting your article "Metabolic constraints drive self-organization of specialized cell groups" for consideration by *eLife*. Your article has been reviewed by two peer reviewers, and the evaluation has been overseen by a Reviewing Editor and Naama Barkai as the Senior Editor. The following individual involved in review of your submission has agreed to reveal their identity: Martin Ackermann (Reviewer #2).

The reviewers have discussed the reviews with one another and the Reviewing Editor has drafted this decision to help you prepare a revised submission.

The two reviewers agree that this is a very interesting manuscript that sheds new light on physiological heterogeneity and possibly even spatially organized metabolic states within an isogenic population of yeast cells. However, both reviewers also raise important questions and are not yet fully convinced by the data presented. We would therefore ask you to consider the following major essential issues:

1) Please provide more arguments, controls and discussion regarding the hypothesized metabolic states, and especially addressing to what extent these might be the consequence of spatial gradients.

2) Can you explain in more detail what you hypothesize to be the physiological/evolutionary reason for the different states?

3) Please expand the statistical analyses.

Please also consider the questions and comments made by the individual reviewers.

*Reviewer #1:*

The results show that within a yeast colony, cells in specific region of the colony show higher activity in the gluconeogenesis pathway (as estimated using Pck1 levels as a biomarker), whereas others do not. Further experiments show that the so-called "light" cells show hallmarks of cells that are actively growing on glucose, whereas dark cells seem to be more starved. A metabolic model is used to try and explain how these two different states could be the result of some cells producing glucose through gluconeogenesis, while others benefit from this resource as some of the glucose (in the form of trehalose) diffused away from the cells with high gluconeogenesis. Deletion of the trehalose transporter Mal11 or the trehalase (Nth1) abolished (or at least reduced) the differences between the two distinct groups of cells.

1) The authors claim that all cells would be expected to have high gluconeogenesis activity, but it is unclear to me whether that is really true. Since (low) glucose is used as a carbon source, it seems possible (or even likely) that some cells may be able to take up sufficient amounts of glucose from the medium, while others (e.g. in more dense and internal regions of the colony) are not. It is of course difficult to avoid such differences in nutrient availability in a colony. However, in principle, it should be possible to obtain less confounded results in a medium that does not contain any glucose (e.g. ethanol + glycerol), which would force all cells to enter gluconeogenesis, unless some cells indeed receive some form of glucose (trehalose) produced by other cells.

2) The test where cells are transferred to medium where gluconeogenesis is really essential (ethanol+glycerol) seems flawed, as in this medium, not only gluconeogenesis but also respiration is required. Furthermore, although I do believe the conclusions coming from this experiment, the control glucose grown cells were pre-grown in liquid instead of on a plate, making them not a true control, since the cells coming from plate will also have to adapt to a liquid medium. You are also missing error bars on graph 2A (i)? Moreover in subsection “Cells organize into spatially restricted, contrary metabolic states within the colony” and Figure 2A, the authors claim a difference in lag, but since only the first 9 hours of growth were measured, I find it difficult to conclude from this graph if it is truly only a lag difference or if the cells perhaps also show differences in growth rate for a much longer time (especially in the gluconeogenic medium).

3) If trehalose uptake is really required for light cells, then why does deletion of MAL11 and NTH1 not abolish the existence of light cells and the specific colony morphology (as seen in Figure 5D)?

*Reviewer #2:*

This manuscript by Varahan and colleagues describes what I think is an intriguing discovery: genetically identical cells in colonies of *Saccharomyces cerevisiae* differentiate into two distinct phenotypic states. The two phenotypes specialise on different metabolic processes and are coupled by the transfer of a metabolite from one cell type to the other. I think the work is both interesting and novel. It is interesting because it shows a clear case of a 'distributed metabolism' in a clonal population: metabolic processes are distributed to different cell types in a clonal population, and the population as a whole exhibits metabolic activity that is based on the coupling between the different sub-processes. The distribution of different metabolic processes to individual cell types could have important consequences. It could alleviate biochemical incompatibilities and allow individual cells to perform metabolic processes faster or more efficiently through specialization. While such metabolic differentiation is thus potentially important, we currently do not understand well how common it is in clonal population.

While a small number of recent studies have focused on this concept (including e.g. Wolfsberg et al. Metab. Engineering 2018 https://www.ncbi.nlm.nih.gov/pubmed/30179665 and Rosenthal et al. *eLife* 2018 https://elifesciences.org/articles/33099), I think this study here describes one of the clearest and most compelling cases. This is also what makes this study novel: the authors use a combination of approaches (including transcriptional reporters, measurements of proteins and metabolite abundances and metabolic flux analysis) to characterize the metabolic states of the two cell types and understand how they are coupled. And they use mathematical modeling to probe the plausibility of their metabolic analysis and to derive predictions that they then address experimentally.

I therefore think this manuscript has the potential to make an important contribution to the field: it addressed a concept that is potentially important but largely unexplored and used a strong combination of experimental and computational approaches that provide substantial insights into how metabolic processes can be distributed across two interacting cell types.

I have two main comments. The first comment is scientific: it did not become clear to me why, on a fundamental level, cells differentiate into two distinct metabolic states, instead of all cells attaining a common intermediate metabolic state. I think this question emerges most clearly in the Discussion, where the authors summarize the main insights. All cells are initially in the same state (gluconeogenesis). They produce trehalose. The trehalose accumulates externally until it reaches a concentration where some cells switch to a different state (glycolytic/high PPP) and start consuming trehalose. Is it understood why there is not a different outcome – that all cells together decrease gluconeogenetic activity over time and all attain the same intermediate metabolic state? I would be very interested in the author's perspective on this issue.

The second comment is about the text: I found the Introduction difficult to understand and am not convinced about how some of the concepts are discussed. For example, in the second paragraph of the Introduction section, almost every sentence contains a statement that I don't understand or that I think does not correspond to how terms are commonly used in the field. I suggest that the authors revisit the Introduction.

---

## [Author Response]

The two reviewers agree that this is a very interesting manuscript that sheds new light on physiological heterogeneity and possibly even spatially organized metabolic states within an isogenic population of yeast cells. However, both reviewers also raise important questions and are not yet fully convinced by the data presented. We would therefore ask you to consider the following major essential issues:1) Please provide more arguments, controls and discussion regarding the hypothesized metabolic states, and especially addressing to what extent these might be the consequence of spatial gradients.

The following arguments, controls and discussions have now been added to the revised manuscript’s Introduction, Results and Discussion sections.

Introduction (new text additions):

The Introduction has substantial inclusions of text, to better explain current approaches and explanations for how new cell states emerge. We then clarify our hypothesis, bringing in our perspective of how possible biochemical constraints might control the emergence of cells with distinct states, before summarizing our findings.

The changes now included in the revised text are included below for convenience:

Introduction paragraph one: “Such phenotypic heterogeneity within groups of clonal cells enables several microbes to persist in fluctuating environments, thereby providing an adaptive benefit for the cell community (Wolf et al., 2005; Thattai et al., 2004).”

Introduction paragraph two: “However, despite striking descriptions on the nature and development of phenotypically heterogeneous states within groups of cells…”

Introduction paragraph three: “In particular, many studies emphasize stochastic gene expression changes that can drive phenotypic heterogeneity (Suel et al., 2007; Ackermann, 2015; Balázsi et al., 2011; Blake et al., 2003). Further, groups of cells can produce…”

Introduction paragraph three: “or support possible co-dependencies (such as commensal or mutual dependencies on shared resources) within the populations (Ackermann, 2015). Such studies now provide insight into why such heterogeneous cell groups might exist, and what the evolutionary benefits might be.”

Introduction paragraph three: “In essence, are there simple chemical or physical constraints, derived from existing biochemical rules and limitations, that explain the emergence and maintenance of heterogeneous phenotypic states of groups of clonal cells in space and over time?”

Introduction paragraph four: “Therefore, if we can understand what these biochemical constraints are, and discern how metabolic states can be altered through these constraints, this may address how genetically identical cells can self-organize into distinct states.“

Introduction paragraph five: “The selective utilization of this resource enables the spontaneous emergence and persistence of cells exhibiting a counter-intuitive metabolic state, with spatial organization. These metabolic constraints create inherent threshold effects, enabling some cells to switch to new metabolic states, while restraining other cells to the original state which produces the resource. This thereby drives the overall self-organization of cells into specialized, spatially ordered communities. Finally, this group of spatially organized, metabolically distinct cells confer a collective growth advantage to the community of cells, rationalizing why such spatial self-organization of cells into distinct metabolic states benefits the cell community.”

Changes/additions in the Results:

Here, we have added several appropriate controls and clarification of text, as well as the results of two major new experiments. These changes are summarized below:

Text clarification, regarding the initial description of the colony morphology: “Currently, the description of such colonies is limited to this external rugose morphology, and does not describe the phenotypic states of cells and/or any spatial organization in the colony.”

Text clarification, on what the expected metabolic state of glucose grown cells are: “The expected metabolic requirements of cells growing in glucose limited conditions are as follows:”

Adding new experimental data (new control, to show that in the low glucose conditions chosen, the extent of gluconeogenesis is uniformly very high), and related text: “Indeed, we first confirmed this second prediction by measuring the amounts of the gluconeogenic enzymes Pck1 (phosphoenolpyruvate carboxykinase) and Fbp1 (fructose-1,6-bisphosphatase), using short-term liquid cultures of cells in either high glucose medium, or the same glucose limited medium we used for colony growth. Expectedly, we observed very high amounts of these gluconeogenic enzymes in cells growing in glucose-limited medium (Figure 1—figure supplement 1A), reiterating that even in well-mixed low glucose medium, cells are in a strongly gluconeogenic state. In order to now explore this hypothesis, of examining the colony to dissect expected metabolic requirements in these conditions, we first designed visual indicators for these metabolic hallmarks of yeast cell growth in low glucose.”

Adding supplemental data, on the distribution of gluconeogenic activity across the entire colony, and not just a section): “This spatial distribution of gluconeogenic activity is shown as a quantitative heat-map histogram overlaid on the entire colony, in Figure 1—figure supplement 2A. Since this observation was based solely on reporter activity, in order to more directly examine this observation […]”

Text clarification on cell growth data: “Expectedly, the dark cells grew robustly and reached significantly higher cell numbers (0D_600_) compared to the light cells in the gluconeogenic condition (Figure 2A).”

Text clarification on interpreting the initial characterization of the two metabolic states seen in the dark or light cells: “While this was an overly simple, and not definitive experiment, counter-intuitively, this result suggested that despite being in a low-glucose environment, the light cells were well suited for growth in high glucose, and therefore might be in a metabolic state suited for growth in glucose. We therefore decided to more systematically investigate this phenomenon.”

Note: we add substantial new data, and better characterize the metabolic states, related to this point. This is also detailed later in the response, and also as a response to a specific reviewer point.

Subsection “Cells organize into spatially restricted, contrary metabolic states within the colony” which explains new data included, further characterizing the cells in the light region: “This spatial restriction of high PPP activity across the colony is also shown as an overlaid quantitative heat-map histogram in Figure 2—figure supplement 2A. Next, we directly addressed the possibility of these light cells exhibiting relatively high PPP activity. For unambiguously testing this, we utilized a stable-isotope based metabolic flux experiment to assess the flux towards PPP in light and dark cells. Light and dark cells isolated from colonies were pulsed with ^13^C-labeled glucose (for ~5 minutes), metabolites extracted, and the incorporation of this carbon label into the late PPP intermediates ribulose-5-phosphate (R-5-P) and sedoheptulose-7-phosphate (S-7-P) was measured by liquid chromatography/mass spectrometry (LC/MS/MS). The relative amounts of these all-carbon labeled PPP intermediates were compared between the two cell types (light or dark). Notably, light cells incorporated significantly higher levels of ^13^C labeled glucose into PPP metabolites compared to the dark cells (Figure 2D, and Figure 2—figure supplement 1B, and see Table 2 for MS parameters), showing that the light cells are in a high PPP activity state.”

Text explanations, for a range of controls done with the model simulations, now included in the manuscript: “A range of resource amounts, growth rates and diffusion of the resource were included in simulations (see Figure 3—figure supplement 1A-C).”

Text clarification: “(a straw-man scenario, since initially in the actual colony all cells were in a dark state, as shown earlier).”

New text addition, to better describe conclusions: “Two simple take-home points emerge from this model, for such spatial distribution of cells in these two metabolic states, in the developing colony.”

Text addition, to better describe the experiment: For this we used stable-isotope labeled trehalose, and measured trehalose uptake, breakdown and utilization.”

Added text to clarify the results: “Indeed, the labeled forms of these metabolites were only above the detection limit in dark cells (Figure 5A and Figure 5—figure supplement 1A).”

Important clarification, on questions regarding the uptake of trehalose: “Note: while Mal11 shows a high affinity for trehalose, *S. cerevisiae* has multiple sugar transporters with reduced affinity for any disaccharide. Therefore, cells lacking *MAL11* may take up trehalose with lower efficiency. In these cells, the ability to break-down trehalose remains intact. More importantly, in colonies of cells lacking trehalase (*Δnth1*), and which therefore cannot efficiently breakdown internal trehalose, […]”

Explaining the results of an included control in the model: “This is also clearly seen in control simulations with a range of resource amounts, and linear switching, as shown in Figure 6—figure supplements 1A-1D. Contrastingly, in the model that successfully simulates the colony development, […]”

Clarification text: “This therefore prompted us to more closely examine the development of actual colonies over time, for these properties.”

Clarification of a result: “This rapid decrease in trehalose after day 4 is despite a steady increase in the total number of cells in the colony (Figure 6—figure supplement 6E).”

Important explanation of a key result, showing the emergence of light cells late during the colony development: “These data also show that the peripheral location of the light cells cannot simply be due to possible greater access to glucose in the medium, since for the first ~4 days, there are no cells in the periphery with high PPP reporter activity. Their emergence is indeed rapid, and switch-like.”

Explanation of our result: “These data suggest a threshold effect, where the controlling resource, trehalose, needs to build up above a certain amount, in order for cells to switch to the contrary, high PPP activity state.

Cells in distinct metabolic states provide a collective growth advantage to the colony New, important data (and new Figure), showing a clear advantage for the community to have these distinct, metabolically specialized cells: “Finally, we wondered if the emergence of light cells in this high PPP activity state might benefit to the community of cells as a whole. To address this, we compared the long term colony expansion of wild-type cells, with colonies of cells lacking the neutral trehalase (*Δnth1*). Cells in the *Δnth1* colonies cannot utilize trehalose to produce glucose, and as shown earlier, will remain in a gluconeogenic state. Therefore, in these *Δnth1* cell colonies light cells will be absent. Strikingly, we observed that as the respective colonies grew over extended periods of time (~21 days), the wild-type colonies spread over a significantly greater area on the plate, while the *Δnth1* colonies were unable to expand as efficiently (Figure 7A and Figure 7B). This shows that the emergence and proliferation of light cells are important for the expansion of the colony. Since the dark cells are required for the emergence and existence of the light cells, collectively, these data suggest how the community uses cells in distinct metabolic states to maximize growth and spatial expansion, possibly to forage for new nutrients.”

Additions in the Discussion:

“A clearer explanation of the concept emerging from our data: “An implicit concept emerging from this study is that of threshold amounts of a controlling or sentinel metabolite that regulates a switch to a new metabolic state. By definition, such a metabolite must be produced by cells present in a certain (original) metabolic state. […] Since the inherent properties of the cells in the distinct states are different, this raises the deeper possibility that these advantages come from physical and chemical properties of the cells, which arise from their distinct metabolic states.”

In response to the reviewing editor’s comment “especially addressing to what extent these might be the consequence of spatial gradients”:

Data from both our experimental data and the computational model strongly suggest that the emergence of the light cells is due to the build-up of a threshold level of trehalose, which is produced by the dark (gluconeogenic) cells. The data clearly shows that (a) if trehalose uptake or (b) utilization is blocked/abolished, practically no cells with high PPP activity emerge in the periphery of the colony (i.e. the light region). These data are now much better illustrated or explained. Second, and importantly, the light cells only emerge (quite suddenly) after 4 days of colony development. This overall phenomenon therefore suggests that this availability of a new resource (trehalose) is necessary for the emergence of new (light) cells. The model effectively implies very short-range reaction-diffusion systems (of the activator- depleted substrate type, since trehalose is locally consumed by the cells that switch to the light state). We have now better clarified these points in the revised discussion. That said, it is possible that some spatial gradients of glucose in the solid agar media still exist, which could contribute to the development of light cells. As the colony develops in the low glucose medium, we consistently see the development of light cells only in the periphery of the colony (Figure 2C). Earlier studies have shown that as the colony develops, the glucose in the agar beneath the colony is completely utilized by the dividing cells and the cells at the periphery have the maximum access to the glucose available on the plate (Regenberg et al., 2016). It is possible that this residual glucose present in the agar plate (areas of the plate where the colonies have not grown) combined with the glucose obtained by the hydrolysis of the taken-up trehalose (which is itself critical for the light cells) might explain why light cells preferentially emerge at the periphery of the colony. This is however mostly speculative, but given the extensive, multiple lines of evidence provided for trehalose being the controlling resource (shown at the level of production, uptake, utilization, and abolishing which there are no light cells) we are very confident about the revised Results and Discussion.

2) Can you explain in more detail what you hypothesize to be the physiological/evolutionary reason for the different states?

Thank you for this suggestion, which prompted us to explore this possibility in greater detail. Instead of only speculating, we now present new data, which directly demonstrate a clear advantage for the colony as a whole to have cells in the light and dark state. This new data is as follows: we find that the presence of the light cells is critical for the spatial expansion of the colony over longer periods of time (measured over ~21 days). This is now included as a new Figure 7A. Notably, colonies of cells that lacks light cells *nth1* deletion strain, which cannot utilize trehalose, and have no light cells with high PPP activity,are not able to expand spatially as effectively (as a colony) compared to the wild-type. This demonstrates that the light cells are important for the overall foraging behavior exhibited by the colony. We also know that dark cells themselves are critical for the emergence of the light cells, since light cells utilize the trehalose produced by the dark cells to fuel their high PPP activity state. Thus, the presence of cells in two distinct metabolic states allows the colony to spatially expand, potentially forage/move to regions of the medium with more access to glucose, and can be thought of as a bet-hedging strategy wherein the cells with different metabolic states (light and dark cells) enable the colony to efficiently expand and forage for nutrients.

Appropriate text has been added to the revised manuscript.

3) Please expand the statistical analyses.

Pertinent statistical analysis has been performed for all applicable data sets in this manuscript. The statistical tests performed along with the corresponding P values have been indicated in the figure legends along with the sample size for all experiments.

Please also consider the questions and comments made by the individual reviewers.Reviewer #1:The results show that within a yeast colony, cells in specific region of the colony show higher activity in the gluconeogenesis pathway (as estimated using Pck1 levels as a biomarker), whereas others do not. Further experiments show that the so-called "light" cells show hallmarks of cells that are actively growing on glucose, whereas dark cells seem to be more starved. A metabolic model is used to try and explain how these two different states could be the result of some cells producing glucose through gluconeogenesis, while others benefit from this resource as some of the glucose (in the form of trehalose) diffused away from the cells with high gluconeogenesis. Deletion of the trehalose transporter Mal11 or the trehalase (Nth1) abolished (or at least reduced) the differences between the two distinct groups of cells.

We thank the reviewer for providing valuable inputs and feedback to help improve the manuscript. We have addressed all the concerns of the reviewer to the best of our abilities, and the included new data and changes are elaborated below.

1) The authors claim that all cells would be expected to have high gluconeogenesis activity, but it is unclear to me whether that is really true. Since (low) glucose is used as a carbon source, it seems possible (or even likely) that some cells may be able to take up sufficient amounts of glucose from the medium, while others (e.g. in more dense and internal regions of the colony) are not. It is of course difficult to avoid such differences in nutrient availability in a colony.

Even in the early stages of development (under low glucose conditions (0.1%)), a majority of cells exhibit gluconeogenesis (Figure 1). This is based on 2 experimental evidences.

First, when we spot the gluconeogenesis reporter strain (*PCK1* reporter) onto a plate containing low glucose (0.1%), within a few hours of growth, the colony exhibits uniform, gluconeogenesis.

**Author response image 1. respfig1:** 12 hr old gluconeogenesis reporter colony (Exposure <5 sec).

This suggests that, at the early stages of colony development, a majority of the cells exhibit gluconeogenesis. This is also evident from Figure 6D in the manuscript (you can see uniform fluorescence in the gluconeogenesis reporter colony imaged at a lower exposure (~1 sec)) for the gluconeogenesis reporter. However, as the colony develops, the gluconeogenesis activity becomes heterogeneous wherein the center of the colony exhibits minimal gluconeogenesis activity (Figure 1—figure supplement 2A), the dark cells exhibit maximum gluconeogenesis activity (Figure 1 (manuscript), Figure 6, Figure 1—figure supplement 2A) and the light cells exhibit almost no gluconeogenesis activity (Figures 1C and 6D).

Second, we also include an additional control, using cells grown in this same low glucose (liquid) medium for a very short time (~6-8 hours, with cells still in log phase), and find that even in a well-mixed system (planktonic cultures), where nutrients are equally distributed for cells unlike a colony, low glucose (0.1%) strongly triggers gluconeogenesis (this we have included as a new supplemental figure, Figure 1—figure supplement 2B). In this experiment, we monitored the amounts of two gluconeogenic enzymes (Pck1 and Fbp1) and compared protein amounts using cells grown in high glucose (2%) or low glucose (0.1%). Low glucose strongly triggered the expression of these enzymes while in high glucose, these enzymes had near-undetectable protein expression.

Appropriate text has been added to the revised manuscript.

Finally, using our model, we simulated the development of colonies using 3 different initial concentrations of gluconeogenic cells A) 95% of cells are gluconeogenic B) 50% of cells are gluconeogenic and C) 10% of the cells are gluconeogenic. We see the robust development of a structured colony with distinct metabolic states (resembling an actual wild-type colony) only when initial percentage of gluconeogenic cells are high (95%). Thus, our model suggests that, for the development of colonies exhibiting distinct metabolic states, a majority of the cells have to be gluconeogenic at the start of colony development. When the majority of the cells are gluconeogenic, the critical resource for the emergence of light cells, trehalose, reaches threshold levels and this enables the switching of dark cells to light cells and this in turn results in the development of a structured colony (Author response image 2). It is possible that threshold levels of trehalose are not attained when we simulate colony development with lesser percentage of gluconeogenic cells (50% or 10%) and hence we do not see a structured colony that resembles a wild-type colony (Author response image 2).

**Author response image 2. respfig2:** A simulation of colony development using the model using different concentrations of gluconeogenic cells A) 95% B) 50% C) 10% during the start of colony development. Black regions represent dark cells and grey regions represent light cells.

However, in principle, it should be possible to obtain less confounded results in a medium that does not contain any glucose (e.g. ethanol + glycerol), which would force all cells to enter gluconeogenesis, unless some cells indeed receive some form of glucose (trehalose) produced by other cells.

We agree with the reviewer’s suggestion that we could have looked at colony development in a medium containing ethanol+glycerol as the sole carbon source. It just so happens that we made all our initial observations in these conditions of low (as opposed to no) glucose entirely serendipitously. We had started with these conditions of low (as opposed to no) glucose only because this is a well-established condition to study colony development in yeast (Granek and Magwene, 2010; Reynolds and Fink, 2001). We subsequently went on to make all our remaining discoveries, using this low glucose condition. Perhaps if we had started our studies with no glucose, we might have made different sets of observations, but the low glucose condition used is a very reasonable growth medium for *S. cerevisiae*, and indeed is a very well established condition (in a range of published studies using yeast to examine gluconeogenesis, the activation of the Snf1 kinase, colony development, etc.). Given that we have backed up our statements with direct metabolite and metabolic flux measurements (which has to our knowledge never been reported from yeast colonies), and more definitively characterize the respective metabolic states, the results are fairly unambiguous. We do now include additional data further comparing the PPP activity (using stable isotope based flux experiments) in the light and dark cells, and this is now included in Figure 2, which establishes these states beyond doubt.

That said, we are now starting to explore what happens to colonies when grown in no glucose (and with ethanol as a sole carbon source). From very preliminary studies, the overall complexity of these colonies seem to be even greater than that in low glucose, and the growth development of these colonies happen over significantly longer durations of time (compared to the low glucose condition). We are currently in the process of determining the metabolic blueprint of these colonies grown in the ethanol+glycerol medium. The morphology exhibited by these colonies are different in these conditions compared to the morphology exhibited by the colonies under low glucose (0.1%). Please refer to Author response image 3 for comparison (and compare to Figure 1 in the manuscript). We believe redoing the entire set of experiments in an entirely different medium composition will be beyond the scope of the current study.

**Author response image 3. respfig3:** 7 day old wild-type gluconeogenic reporter colonies (*PCK1* reporter) grown in a medium containing ethanol+glycerol as the sole carbon source.

2) The test where cells are transferred to medium where gluconeogenesis is really essential (ethanol+glycerol) seems flawed, as in this medium, not only gluconeogenesis but also respiration is required. Furthermore, although I do believe the conclusions coming from this experiment, the control glucose grown cells were pre-grown in liquid instead of on a plate, making them not a true control, since the cells coming from plate will also have to adapt to a liquid medium. You are also missing error bars on graph 2A (i)? Moreover in subsection “Cells organize into spatially restricted, contrary metabolic states within the colony” and Figure 2A, the authors claim a difference in lag, but since only the first 9 hours of growth were measured, I find it difficult to conclude from this graph if it is truly only a lag difference or if the cells perhaps also show differences in growth rate for a much longer time (especially in the gluconeogenic medium).

We agree with the author that respiration is essential for cells growing in a medium containing ethanol+glycerol as the sole carbon source. However, we do not draw any strong conclusions with this experiment. In fact, we did this only as a preliminary experiment to determine the existence of different metabolic state within the same community (and not to explore the need for respiration). The key take home message from the this experiment was only that dark cells that exhibit a gluconeogenic state are more adapted to growth in a medium containing ethanol+glycerol compared to the light cells which exhibit an alternate metabolic state. That is it. Similarly, the light cells are more adapted to grow in a medium containing glucose compared to the dark cells. This is in fact what prompted us to better characterize the light cells, and led us to perform far more decisive experiments. The reporters for high PPP activity (Figure 2B) were revelatory, suggesting that these cells had an unexpected, high PPP activity state. However, in the earlier version, we did not have direct metabolic evidence for this. Therefore, we now include a new, stable-isotope based metabolic flux experiment, providing ^13^C glucose to light and dark cells for ~5min, and directly measuring flux to the late PPP metabolites (ribose-5-phosphate, and sedoheptulose-7-phosphate). We find that light cells have significantly higher flux into PPP (as measured by these newly synthesized metabolites). Further, we also show that light cells, synthesize more nucleotides compared to the dark cells, with new nucleotide synthesis being a canonical end point of high PPP activity. We include this new data, directly measuring PPP flux, in the new figure 2D, as well as Figure 2—figure supplement 1B. The PPP reporter data remains as Figure 2D, while the nucleotide data are now in Figure 2E. These data now unambiguously demonstrate that the light cells have substantially more PPP activity than adjacent dark cells.

We have made appropriate text revisions, to clarify this point.

Furthermore, although I do believe the conclusions coming from this experiment, the control glucose grown cells were pre-grown in liquid instead of on plate, making them not a true control, since the cells coming from plate will also have to adapt to a liquid medium.

We agree with the reviewer that cells coming from a plate will take time to adapt to a liquid medium. However, cells grown in high glucose (although planktonic) merely serve as a control to show light cells behave like cells growing in glucose rich conditions. Also, see response to the previous point (where we now directly demonstrate that the light cells have substantially higher PPP activity than the dark cells)

You are also missing error bars on graph 2A (i)?

We thank the reviewer for pointing this out. Error bars have now been added to this graph and they represent standard deviation.

Moreover in “Cells organize into spatially restricted, contrary metabolic states within the colony” and Figure 2A, the authors claim a difference in lag, but since only the first 9 hours of growth were measured.

We agree with the reviewer and the text has been modified as follows:

“Expectedly, the dark cells grew robustly and reached significantly higher cell numbers (0D_600_) compared to the light cells in the gluconeogenic condition (Figure 2A).*”*

3) If trehalose uptake is really required for light cells, then why does deletion of MAL11 and NTH1 not abolish the existence of light cells and the specific colony morphology (as seen in Figure 5D)?

As established in this manuscript, one of the hallmarks of light cells is their high PPP activity. Figure 5B and 5C clearly shows that *∆nth1* strain, which cannot efficiently break-down and utilize trehalose, does not have *any* detectable light cells compared to the wild-type. The *∆nth1* strain harboring the PPP reporter plasmid shows *no* fluorescence suggesting that ability to break down trehalose is critical for the emergence of light cells.

Separately, the *∆mal11* strain has significantly reduced populations of light cells compared to the wild-type as evident from Figure 5B and 5C. This could be due to contribution of other low–affinity trehalose transporters in the absence of Mal11/ATG1. Several studies have shown that *S. cerevisiae* possess low affinity trehalose transport activity in the absence of the high affinity transporter Mal11/ATG1 (e.g. Stambuk et al., 1996, Stambuk et al., 1998). *S. cerevisiae* is known to harbor several α-glucoside transporters in its genome (Chow et al., 1996, Lagunas 1993) and it is likely that any one of these could contribute towards the low-affinity transport of trehalose in a *∆mal11* strain.

Finally, the deletion of *MAL11* and *NTH1* dramatically decreases the emergence of the light cells which are localized to the periphery of the colony (Figure 5C). However, the overall apparent colony morphology seems unaffected, as the ‘rugose’ morphology is predominantly contributed by the dark cells (gluconeogenic) and not light cells. As shown in Figure 5D, the population of dark cells remain unaffected even after the deletion of the trehalose uptake and utilization machinery.

Appropriate text has been added to the revised manuscript.

Reviewer #2:[…] I have two main comments. The first comment is scientific: it did not become clear to me why, on a fundamental level, cells differentiate into two distinct metabolic states, instead of all cells attaining a common intermediate metabolic state. I think this question emerges most clearly in the Discussion, where the authors summarize the main insights. All cells are initially in the same state (gluconeogenesis). They produce trehalose. The trehalose accumulates externally until it reaches a concentration where some cells switch to a different state (glycolytic/high PPP) and start consuming trehalose. Is it understood why there is not a different outcome – that all cells together decrease gluconeogenetic activity over time and all attain the same intermediate metabolic state? I would be very interested in the author's perspective on this issue.

We thank the reviewer for providing valuable inputs and feedback to help improve the manuscript. We have addressed all the concerns of the reviewer to the best of our abilities.

We thank the reviewer for this specific suggestion, although this was not the initial objective of our manuscript. Here we only focus on how cells in the two states can emerge and organize spatially. However, this query prompted us to further explore what the advantages of these two states might be, and we obtain striking data in this regard. Our data demonstrates that the existence of light cells is critical for the expansion of the colony (now shown as Figure 7A). Colonies from cells that lacks light cells (the *nth1* (trehalase enzyme) deletion strain) cannot expand as effectively over space, as compared to the wild-type. This suggests that light cells are important for the foraging behavior exhibited by the colony, potentially to expand to new areas with better access to food. We earlier demonstrate that dark cells are critical for the emergence of the light cells, since light cells utilize the trehalose produced by the dark cells to fuel their high PPP metabolic state (Figure 6). Thus, perhaps the presence of two distinct metabolic states is a bet-hedging strategy wherein the cells with different metabolic states (light and dark cells) enable the colony to efficiently expand and forage for nutrients.

Without making overstatements, we include appropriate text along with this new data to the revised manuscript.

The second comment is about the text: I found the Introduction difficult to understand and am not convinced about how some of the concepts are discussed. For example, in the second paragraph of the Introduction section, almost every sentence contains a statement that I don't understand or that I think does not correspond to how terms are commonly used in the field. I suggest that the authors revisit the Introduction.

We have made substantial changes and additions to the manuscript Introduction and Discussion. In the earlier submission, we perhaps lost clarity for brevity. This revised version should be far clearer. We also apologize for any inadvertent errors in the usage of terms, since all of us come from slightly different fields. Our objective was to bring a more direct biochemical perspective to this field, where the data is from different experimental approaches.

We have made several changes and clarifications to the text, in the Introduction. Do refer to the general responses for details of all the changes made.